# Combinatorial Multi-armed Bandits: Arm Selection via Group Testing

**Arpan Mukherjee**\*      *a.mukherjee@imperial.ac.uk*
*Department of Electrical and Electronic Engineering*
*Imperial College London*

**Shashanka Ubaru**      *shashanka.ubaru@ibm.com*
*IBM Research*

**Keerthiram Murugesan**      *keerthiram.murugesan@ibm.com*
*IBM Research*

**Karthikeyan Shanmugam**$^{\dagger}$      *karthikeyanshanmugam88@gmail.com*
*IBM Research*

**Ali Tajer**      *tajer@ecse.rpi.edu*
*Department of Electrical, Computer and Systems Engineering*
*Rensselaer Polytechnic Institute*

**Reviewed on OpenReview:** https://openreview.net/forum?id=Mq59rTnIfE

## Abstract

This paper addresses the problem of combinatorial multi-armed bandits with semi-bandit feedback and a cardinality constraint on the size of the super-arm. Existing algorithms for solving this problem typically involve two key sub-routines: (1) a *parameter estimation* routine that sequentially estimates a set of base-arm parameters, and (2) a *super-arm selection* policy for selecting a subset of base arms deemed optimal based on these parameters. State-of-the-art algorithms assume access to an *exact* oracle for super-arm selection with unbounded computational power. At each instance, this oracle evaluates a list of score functions, the number of which grows as low as linearly and as high as exponentially with the number of arms. This can be prohibitive in the regime of a large number of arms. This paper introduces a novel realistic alternative to the perfect oracle. This algorithm uses a combination of *group-testing* for selecting the super arms and *quantized* Thompson sampling for parameter estimation. Under a general separability assumption on the reward function, the proposed algorithm reduces the complexity of the super-arm-selection oracle to be *logarithmic* in the number of base arms while achieving the same regret order as the state-of-the-art algorithms that use exact oracles. This translates to *at least an exponential* reduction in complexity compared to the oracle-based approaches.

## 1 Introduction

The combinatorial multi-armed bandit (CMAB) problem is a generalization of the stochastic multi-armed bandit problem, in which there is a set of *base* arms and a learner selects a *subset* of them at each round. Such

---

\*This research was partially conducted during the author's internship at IBM Research.

$^{\dagger}$The author's contribution was when the author was a part of IBM Research. The author is currently with Google Deepmind India, Bangalore.

sets of base arms are called *super-arms*, and the set of all possible super-arms constitutes the action space of the learner (Chen et al., 2013; Combes et al., 2015; Chen et al., 2016a;b; Wang & Chen, 2017; Perrault, 2022).

**Bandit versus semi-bandit feedback.** CMABs can be broadly divided into two settings according to the level of feedback a learner receives in response to its actions: the *bandit* and the *semi-bandit* feedback settings. In the bandit feedback setting, the learner pulls a super-arm and observes the aggregate reward value generated by the selected super-arm Nie et al. (2022); Jia et al. (2019). On the other hand, in the semi-bandit feedback setting, in addition to the aggregate reward, the learner has access to a set of stochastic observations generated by the individual arms that constitute the selected super arm (Chen et al., 2016a; Wang & Chen, 2018). This paper focuses on semi-bandit feedback and aims to minimize the average cumulative regret in CMABs under this feedback model. The CMAB model is assumed to belong to the class of Bernoulli bandits.

**UCB versus Thompson sampling.** Regret minimization algorithms for CMABs with semi-bandit feedback consist of two key sub-routines: an estimation routine and a super-arm selection routine. The estimation routine aims to form reliable estimates of the unknown parameters of the base arms. The super-arm selection routine specifies the sequential selection of the super-arms over time. Super-arm selections rely on the estimates formed by the estimation routine, and there is a wide range of arm-selection rules based on the upper confidence bound (UCB) principle (Chen et al., 2013; Kveton et al., 2015; Combes et al., 2015; Chen et al., 2016a) or Thompson sampling (TS) (Wang & Chen, 2018; Perrault et al., 2021). Recent studies demonstrate that the TS-based approaches are more efficient and empirically outperform the UCB-based counterparts. Specifically, the combinatorial Thompson sampling (CTS) algorithm in (Wang & Chen, 2018) adopts a posterior sampling estimator for the bandit mean values, and uses an oracle that perfectly determines the set of super-arms that are optimal for the estimated means. Under such access to an *exact* oracle, the studies in (Wang & Chen, 2018) and (Perrault et al., 2021) establish that the CTS algorithm achieves an order-wise optimal regret of $O\left(\frac{m}{\Delta}\log T\right)$, where $m$ denotes the number of base arms, $T$ is the horizon, and $\Delta$ specifies the minimum expected reward gap between an optimal super-arm and any other non-optimal super-arm. (Merlis & Mannor, 2019; Liu et al., 2022) investigate the CMAB problem in the batched setting. Furthermore, (Merlis & Mannor, 2020) provides tight lower bounds for CMABs.

**Oracle complexity.** Accessing an exact oracle is often computationally prohibitive. In this paper, our objective is to alleviate the *oracle complexity* of existing methods. This is motivated by the fact that black-box function evaluations can be expensive, and hence, it is imperative to minimize the number of black-box queries to the oracle. It is noteworthy that there exist *approximate* alternatives to the exact oracle, which require a polynomial complexity in the number of base arms. While offering an improvement in complexity, polynomial complexity can still be excessive, and more importantly, approximate solutions can result in *linear* cumulative regret. Examples of reward functions facing such issues include submodular reward functions (Krause & Golovin, 2014) and reward functions modeled as the output of a neural network (Hwang et al., 2023). Therefore, to avoid linear regret, CTS has to inevitably rely on an exact oracle, the computational complexity of which, in general, grows exponentially with the number of base arms.

**Group testing.** Group testing (GT) is an efficient approach for solving large-scale combinatorial search problems (Dorfman, 1943; Du et al., 2000). The basic premise in group testing (GT) is that a small sub-population (size $K$) of a large body (size $m \gg K$) possesses a particular property (e.g., being defective), and the objective of group testing is to identify it without individually testing all members. To avoid individual tests, the population members are *pooled* into groups, and the group is tested as a whole. The majority of tests are expected to return negative results, i.e., most groups do not have a member with the desired property. This clears the entire group, significantly saving the number of tests administered.

The number of tests required to identify defective items varies widely depending on the settings (see (Aldridge et al., 2019) for a review). Under both noiseless as well as noisy test outcomes (with a bound on the number of noisy measurements), when $K = O(m^\alpha)$ where $\alpha \in (0, 1/3)$, only $O(K^2 \log m)$ tests are sufficient to recover the defective subset perfectly (zero-error criterion) (Hwang & T. Sós, 1987; Du et al., 2000; Chan et al., 2011). Under a vanishing error criterion, the number of tests can be reduced to $O(K \log m)$ (Zhigljavsky, 2003; Gilbert et al., 2012) (partial recovery). Furthermore, GT schemes can be classified into adaptive and non-adaptive methods. In non-adaptive group testing, all the tests are conducted simultaneously. In contrast,

in adaptive group testing, the tests are divided into stages, and the tests for a particular stage are decided based on the outcomes of the previous stage. Adaptive group testing has been shown to significantly reduce the number of tests, requiring only $O(K \log m + m)$ tests for exact recovery (Hwang, 1975; De Bonis et al., 2005).

Different variants of group testing have also been proposed in the literature (Du et al., 2000; Du D, 2006; D'yachkov, 2014). These include threshold group testing (Damaschke, 2006), where a test result is positive if the number of defective items in the pool is above a threshold; quantitative or additive group testing (D'yachkov, 2014; Du et al., 2000), where the test output is the number of defective items in the pool; probabilistic group testing (Cheraghchi et al., 2011), where we wish to recover the defective items with high probability; graph-constrained group testing (Sihag et al., 2021), where there are constraints how items can be grouped; and semi-quantitative group testing (Emad & Milenkovic, 2014; Cheraghchi et al., 2021), where the (additive) test outputs are quantized into a fixed set of thresholds. GT has also been adopted to solve large-scale learning problems, such as feature selection (Zhou et al., 2014), extreme classification (Ubaru & Mazumdar, 2017; Ubaru et al., 2020), and data valuation (Jia et al., 2019).

**Contributions.** In this paper, we leverage GT to dispense with the assumption of exact oracle access for the CTS algorithm. This results in an *exponential reduction* in the oracle complexity without compromising the achievable regret. Specifically, we devise the **G**roup **T**esting + **Q**uantized **T**hompson **S**ampling (GT+QTS) algorithm, which under a mild probabilistic assumption on the separability of the reward function (Assumption 6), will have exponentially lower complexity compared to an exact oracle. Reducing oracle complexity is a fundamental challenge with significant practical relevance; a detailed discussion can be found in Appendix A. GT+QTS has two key innovations compared to the existing algorithms. First, the complexity reduction is enabled by GT, the success of which fundamentally relies on separability assumptions, lacking which we may face sub-optimal (linear) regret. To address this, as a second contribution, we devise a *quantization* scheme that ensures the probabilistic separability of the reward function. We show that the GT-based oracle requires only $O(\log m)$ black-box queries to discern the optimal set of arms in each round. Furthermore, we show that the GT+QTS algorithm preserves the optimal regret order of $O\left(\frac{m}{\Delta} \log T\right)$ while providing an exponential reduction in the oracle complexity.

**Related works.** We provide an overview of the most closely related studies to the scope of this paper. The theoretical analysis of the TS-based approaches for MABs was first provided in (Kaufmann et al., 2012; Agrawal & Goyal, 2012). These results were later improved in (Agrawal & Goyal, 2013) and extended to a general action space and feedback in (Gopalan et al., 2014). The CMAB problem is studied under different settings in (Chen et al., 2013; Combes et al., 2015; Chen et al., 2016b;a). The TS-based approach to CMAB is investigated for top-$K$ CMAB in (Komiyama et al., 2015), analyzed for contextual CMAB in (Wen et al., 2015), and studied under the Bayesian regret metric by Russo & Van Roy (2016). Furthermore, CMAB has been investigated in the full-bandit feedback setting in Nie et al. (2022).

The study closest to the scope of this paper is Wang & Chen (2018), which analyzes the CTS algorithm to solve combinatorial semi-bandits under a Bernoulli model and a Beta prior distribution for the belief parameters. It establishes that the CTS algorithm asymptotically achieves the optimal regret. Another related study is by Perrault et al. (2021), which presents a tighter regret bound for the Beta prior. A similar optimal regret analysis is established for multivariate sub-Gaussian outcomes using Gaussian priors.

## 2 Combinatorial Bandits

**Setting.** Similarly to the canonical models in (Wang & Chen, 2018; Perrault et al., 2021), we consider a CMAB setting with $m$ arms, and define the set $[m] := \{1, \cdots, m\}$. Each arm $i \in [m]$ is associated with an independent Bernoulli distribution with an *unknown* mean $\mu_i$. We denote the vector of *unknown* mean values by $\boldsymbol{\mu} := [\mu_1, \cdots, \mu_m]$. Sequentially over time, the learner selects subsets of arms, which we refer to as *super-arms*. The super-arm selected at time $t$ is denoted by $\mathcal{S}(t) \in \mathcal{I}$, where $\mathcal{I} = 2^{[m]}$ specifies the set of super-arms. We consider the semi-bandit feedback model, wherein, at each time $t$, upon pulling a super-arm

$\mathcal{S}(t)$, the learner observes a feedback

$$Q(t) := \{X_i(t) : i \in \mathcal{S}(t)\} , \tag{1}$$

where $X_i(t)$ denotes a random observation from arm $i \in [m]$, i.e., $X_i(t) \sim \mathrm{Bern}(\mu_i)$. In addition to the feedback $Q(t)$, based on the super-arm selected at time $t$, the learner gains a reward $R(t)$. The average reward $\mathbb{E}[R(t)]$ is assumed to depend only on the mean values of the arms $i \in \mathcal{S}(t)$. To formalize this, we assume that there exists a function $r : \mathcal{I} \times [0,1]^m \mapsto \mathbb{R}$, such that

$$\mathbb{E}[R(t)] = r(\mathcal{S}(t) \, ; \, \boldsymbol{\mu}) , \tag{2}$$

where the expectation is with respect to the measure induced by the distributions of arms $i \in \mathcal{S}(t)$. Function $r$ is assumed to be *unknown*, and the learner only has *black-box* access to it, i.e., for any $\boldsymbol{\theta} \in [0,1]^m$ and $\mathcal{S} \in \mathcal{I}$, the learner queries the black-box and obtains the reward evaluation $r(\mathcal{S} \, ; \, \boldsymbol{\theta})$. For any $\boldsymbol{\theta} \in [0,1]^m$, we define the optimal super-arm associated with $\boldsymbol{\theta}$ as the permissible set with the largest reward, i.e.,

$$\mathcal{S}^\star(\boldsymbol{\theta}) := \arg\max_{\mathcal{S} \in \mathcal{I}} r(\mathcal{S} \, ; \, \boldsymbol{\theta}) . \tag{3}$$

If there are multiple optimal super-arms, we randomly select one of them. For a given $\boldsymbol{\theta}$ and any set $\mathcal{S} \in \mathcal{I}$, we define the sub-optimality with respect to $\mathcal{S}^\star(\boldsymbol{\theta})$ by

$$\Delta(\mathcal{S}, \boldsymbol{\theta}) := r(\mathcal{S}^\star(\boldsymbol{\theta}) \, ; \, \boldsymbol{\theta}) - r(\mathcal{S} \, ; \, \boldsymbol{\theta}) . \tag{4}$$

Accordingly, we define the minimal and maximal sub-optimality gaps for any parameter $\boldsymbol{\theta} \in [0,1]^m$ as

$$\Delta_{\min}(\boldsymbol{\theta}) := \min_{\mathcal{S} \in \mathcal{I} : \Delta(\mathcal{S}, \boldsymbol{\theta}) > 0} \Delta(\mathcal{S}, \boldsymbol{\theta}) , \quad \text{and} \quad \Delta_{\max}(\boldsymbol{\theta}) := \max_{\mathcal{S} \in \mathcal{I}} \Delta(\mathcal{S}, \boldsymbol{\theta}) . \tag{5}$$

The learner's objective is to minimize the *average* cumulative regret $\mathfrak{R}(T)$, which is defined as

$$\mathfrak{R}(T) := \sum_{t=1}^{T} \mathbb{E}[\Delta(\mathcal{S}(t), \boldsymbol{\mu})] , \tag{6}$$

where the expectation is taken with respect to the measure induced by the learner's interaction with the bandit instance. For any set $\mathcal{S} \subseteq [m]$ and $\boldsymbol{\theta} \in [0,1]^m$, we define $\boldsymbol{\theta}_{\mathcal{S}}$ as the vector, whose entries are equal to $\boldsymbol{\theta}$ for every $i \in \mathcal{S}$, and 0 otherwise.

**Assumptions.** We start by discussing some of the commonly used assumptions in the CMAB literature on the reward function $r$ (Wang & Chen, 2018; Perrault et al., 2021). Then, we will discuss how to relax some of the idealized assumptions in the literature. Specifically, the existing studies relevant to this work assume access to an exact oracle that can perfectly solve the problem in (3), i.e., it identifies the optimal super-arm $\mathcal{S}^\star(\boldsymbol{\theta})$ for any parameter $\boldsymbol{\theta} \in [0,1]^m$. In this paper, we relax this assumption and replace the oracle with a procedure with only soft (probabilistic) guarantees for solving (3). We begin with the following common assumption in CTS-based approaches for CMAB; see (Wang & Chen, 2018; Perrault et al., 2021).

**Assumption 1.** *The expected reward of a super-arm $\mathcal{S} \in \mathcal{I}$ depends only on the mean values of the base arms in $\mathcal{S}$.*

We note that some studies on the confidence interval-based methods have relaxed this assumption (Chen et al., 2016a). In the context of CTS, relaxing this assumption presents several technical challenges. Specifically, a TS-based approach at each step samples the super-arm that maximizes the reward function based on posterior *mean* estimates. However, for rewards, which depend on the arm distributions (and not just the mean values), we need estimates for the distributions. This calls for a separate algorithm design. Our following assumption quantifies the smoothness of the reward function.

**Assumption 2** (Lipschitz continuity)**.** *The reward function is globally $B$-Lipschitz in $\boldsymbol{\theta}$. More specifically, for any $\mathcal{S} \in \mathcal{I}$ and for any $\boldsymbol{\theta}, \boldsymbol{\theta}' \in [0,1]^m$, the reward function satisfies $|r(\mathcal{S} \, ; \, \boldsymbol{\theta}) - r(\mathcal{S} \, ; \, \boldsymbol{\theta}')| \leq B\|\boldsymbol{\theta}_{\mathcal{S}} - \boldsymbol{\theta}'_{\mathcal{S}}\|_1$ for some universal constant $B \in \mathbb{R}_+$.*

Next, we specify assumptions on the variations of the reward function with respect to $\mathcal{S}$. We adopt a common monotonicity assumption based on which adding arms to any super-arm will not decrease the reward.

**Assumption 3** (Reward monotonicity). *The reward function $r$ is monotone and increasing in $\mathcal{S}$, i.e., for any $\mathcal{S}_1 \subseteq \mathcal{S}_2 \subseteq [m]$ we have $r(\mathcal{S}_1 \; ; \; \boldsymbol{\theta}) \le r(\mathcal{S}_2 \; ; \; \boldsymbol{\theta}), \forall \boldsymbol{\theta} \in [0,1]^m$.*

Without any constraint on the cardinality of the optimal set, the monotonicity assumption implies that the optimal super-arm is $[m]$. To avoid this, we impose that the cardinality of the optimal super-arm $|\mathcal{S}| \le K \in [m]$. Besides the above standard CMAB assumptions, we also adopt three more assumptions pertinent to dispensing with access to the exact oracle that solves (3) and designing an efficient probabilistic alternative. The following two assumptions are needed for determining the number of tests in our GT procedure.

**Assumption 4** (Bounded reward). *For any set $\mathcal{S} \in \mathcal{I}$ and any $\boldsymbol{\theta} \in [0,1]^m$, we assume that the reward function satisfies $r(\mathcal{S} \; ; \; \boldsymbol{\theta}) \in [0, M]$, where $M$ is known.*

Next, we introduce a probabilistic assumption on the distribution of the minimum gap of the bandit instances. This assumption is critical for facilitating an exponential reduction in oracle complexity. Furthermore, this assumption covers the case where a lower bound on the minimum gap of the class of instances is known/assumed to be known, which is a common occurrence in many applications such as principal component analysis (minimum singular value gap requirement for iterative partial SVD algorithms Musco & Musco (2015)), topological data analysis (minimum gap requirement for Betti number estimation Apers et al. (2023)), and others.

**Assumption 5.** *The probability of distribution of the minimum gap $\Delta_{\min}(\boldsymbol{\mu})$ is known, and its cumulative distribution function (CDF) is denoted by $\mathbb{F}_{\boldsymbol{\mu}}$.*

The next assumption states that augmenting any subset of arms with an optimal arm results in higher reward gain than augmenting with a non-optimal arm.

**Assumption 6** (Separable reward). *For any parameter $\boldsymbol{\theta} \in [0,1]^m$, any optimal arm $s \in \mathcal{S}^\star(\boldsymbol{\theta})$, any sub-optimal arm $\tilde{s} \notin \mathcal{S}^\star(\boldsymbol{\theta})$, and any set $\mathcal{S} \subset [m] \setminus \{s, \tilde{s}\}$, we have[1]:*

$$r(\mathcal{S} \cup \{s\} \; ; \; \boldsymbol{\theta}) - r(\mathcal{S} \cup \{\tilde{s}\} \; ; \; \boldsymbol{\theta}) > 0 \; . \tag{7}$$

In many practical settings, Assumption 5 is an artifact of designing the experiments and representing them by bandit arms. Specifically, in real-world settings, when two experiments are deemed to have sufficiently close rewards or utilities, they are effectively considered the same experiment. From this perspective, $\Delta_{\min}$ can be viewed as the minimum separation of rewards based on which we consider the experiments sufficiently distinct to warrant representing them by distinct arms. So, this is an application-specific constant, and depending on the underlying application of interest and what the arms represent, it can be set by the domain expert. Furthermore, in various applications, feedback or utility values are inherently quantized, leading to a **natural lower bound** on the possible difference between two super-arm utilities. For example, consider the case of a recommendation system where the learner aims to suggest content based on user feedback, balancing exploration (new content) with the avoidance of low-quality recommendations. User feedback in such systems is typically **discrete** (e.g., ratings in the set $\{1, 2, 3, 4, 5\}$). In this case, the smallest possible difference in utility between any two super-arms is lower bounded by 1, providing a concrete and known lower bound on $\Delta_{\min}$. Furthermore, several commonly used set-valued functions naturally satisfy the separability assumption, e.g., linear rewards, i.e., $r(\mathcal{S} \; ; \; \boldsymbol{\theta}) = \sum_{i \in \mathcal{S}} \theta_i$, information measure-based functions such as mutual information and $f$-divergence (Zhou et al., 2014; Nguyen et al., 2010). Furthermore, we show that a two-layer neural network (NN) also satisfies the separability assumption (see Theorem 2, Appendix E).

## 3 Algorithm: GT + Quantized TS

In this section, we provide details of the GT+QTS algorithm, the objective of which is to minimize the average cumulative regret defined in (6). This algorithm has two central sub-routines. The first sub-routine is

---

[1]In the case of multiple optimal super-arms, $s$ belongs to the union of optimal arms, and $\tilde{s}$ does not belong to it.

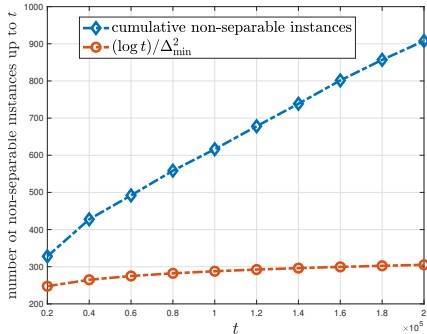

Figure 1: Cumulative number of times that the instance $\boldsymbol{\theta}(t)$ is non-separable.

an estimation process that computes estimates for the base arm means. The second sub-routine is a procedure that, at each round, sequentially determines an optimal super-arm to be pulled based on the current base arms' mean estimates. These procedures are discussed next.

## 3.1 TS-based Estimator

We consider a TS-based approach, where the estimates of the mean values are generated by sampling from a posterior distribution. We adopt a beta distribution to generate the posteriors. A beta posterior naturally comes up as the conjugate distribution assuming uniform priors on the mean values of the base arms. We denote the distribution associated with arm $i \in [m]$ at time $t$ by $\mathsf{Beta}(a_i(t), b_i(t))$. We initialize, for $t = 0$, $a_i(0) = b_i(0) = 1$ for all arms, in which case the beta distribution reduces to a uniform distribution. Subsequently, for each time $t \in \mathbb{N}$, a super-arm $\mathcal{S}(t)$ is selected, and we receive the feedback $Q(t)$. Based on the feedback, we update the prior distribution of each base arm by updating $a_i(t)$ and $b_i(t)$. Furthermore, recall that $X_i(t)$ denotes the feedback from the base arm $i \in \mathcal{S}(t)$. We draw a sample $Y_i(t) \sim \mathsf{Bern}(X_i(t))$, and update the posterior distribution as follows.

$$a_i(t+1) \;=\; a_i(t) + Y_i(t) \;, \tag{8}$$
$$b_i(t+1) \;=\; b_i(t) - Y_i(t) + 1 \;. \tag{9}$$

Finally, our estimate for $\boldsymbol{\mu}$ at time $t$ is a random sample from the beta distribution with parameters specified in (8)-(9), i.e., we generate the posterior estimate $\boldsymbol{\theta}(t+1)$ according to $\theta_i(t+1) \sim \mathsf{Beta}(a_i(t+1), b_i(t+1))$.

## 3.2 GT-based Arm Selection

We design a GT-based procedure to select the optimal super-arm in each round. The nature of this procedure is probabilistic, and it is designed to find the optimal super-arm with a high probability.

GT involves pooling together several arms and performing a test on the pooled set. Tests are repeated by selecting and pooling different subsets of arms for each test. When we have $\ell$ tests, the pooling process can be characterized by a test matrix $\mathbf{A} \in \{0,1\}^{\ell \times m}$, where row $j \in [\ell]$ specifies the arms that are included in test $j$. Specifically, $A_{j,i} = 1$ if the arm $i \in [m]$ is contained in test $j \in [\ell]$, and otherwise $A_{j,i} = 0$. For each test $j \in [\ell]$, we design a function $\rho_j : 2^{[m]} \times [0,1]^m \mapsto [0,M]$, that assigns score to the outcome of test $j$. Next, based on these test scores, we assign a grade to each arm that specifies whether the arm is likely to be in the optimal super-arm or not. This grade assignment is formalized by a decoding mechanism specified by the function $\phi_i : [0,M]^\ell \mapsto \mathbb{R}$, which generates the arms' grades. Subsequently, a candidate super-arm is selected as the set of arms with the top $K$ grades.

**Group-testing oracle (GTO).** Next, we describe our GT encoding and decoding mechanisms. To lay context, we first describe a naïve adaptation of the GT approach in (Zhou et al., 2014). It was designed for ranking and can be used to find the optimal super-arm at each step. We then describe a shortcoming of this naïve approach and modify it to replace the exact oracle used by CTS.

---

**Algorithm 1:** GT+QTS Algorithm

---

**Input :** Cardinality constraint $K$, # rounds $T$

**1** *Initialize* $t = 1$, $a_i(t) = 1$, $b_i(t) = 1$ for all $i \in [m]$

**2 for** $t = 1 \ldots T$ **do**

**3** $\quad$ Draw a sample $\theta_i(t) \sim \mathsf{Beta}(a_t(t), b_i(t))$ for every arm $i \in [m]$, and form $\boldsymbol{\theta}(t)$

**4** $\quad$ Play the super-arm $\mathcal{S}(t)$ returned by $\mathrm{Oracle}(\boldsymbol{\theta}(t))$ (Algorithm 2)

**5** $\quad$ Obtain the observations $Q(t)$

**6** $\quad$ Update $a_i(t+1)$ & $b_i(t+1)$ according to (8) (9)

**7 end**

---

*Naïve GT approach.* We adopt a randomized testing mechanism, in which each arm $j \in [m]$ is included in the test by flipping a coin. Specifically, arm $i \in [m]$ is included in test $j \in [\ell]$ based on a Bernoulli random variable $A_{j,i} \sim \mathsf{Bern}(p)$ such that arm $i$ is included in the test if $A_{j,i} = 1$. Probability $p$ is a design parameter to be chosen later. For designing the decoder, at each time $t \in \mathbb{N}$, we set the scoring function of test $j$, i.e., $\rho_j$, to be an evaluation of the average reward function at the current estimates of the mean values, i.e., $\rho_j(t) := r(\mathbf{A}_j \; ; \; \boldsymbol{\theta}(t))$, where $\mathbf{A}_j$ denotes the $j^{\mathsf{th}}$ row of the test matrix $\mathbf{A}$. Based on the test scores $\boldsymbol{\rho}(t) := (\rho_1(t), \cdots, \rho_\ell(t))$, we define the arm grading function $\boldsymbol{\phi}(t) := (\phi_1(t), \cdots, \phi_m(t))$ as follows.

$$\boldsymbol{\phi}(t) \; = \; \mathbf{A}^\top \boldsymbol{\rho}(t) \; . \tag{10}$$

For each arm $i \in [m]$, the GT decoder in (10) considers the tests $j \in [\ell]$ which contain $i$, and adds up the scores due to these tests to form an aggregate grade for each arm $i$. If an arm $i \in [m]$ is contained in multiple tests with high scores, and the resulting aggregate score is large, it is highly likely that the arm $i$ is responsible for the high scores assigned to the tests. Hence, arm $i$ is a more likely candidate to be one of the arms in the optimal super-arm. Hence, the arms with the top-$K$ grades are selected as candidates for the optimal super-arm to be pulled at time $t$.

*Separability.* The naïve GT mechanism faces a delicate shortcoming; for the GT to work, the reward function $r(\cdot; \boldsymbol{\theta}(t))$ must satisfy a $C$-separability assumption, which is stronger than Assumption 6. Specifically, under $C$-separability, any two arms $s \in \mathcal{S}^\star(\boldsymbol{\theta}(t))$ and $\tilde{s} \notin \mathcal{S}^\star(\boldsymbol{\theta}(t))$, and any set $\mathcal{S} \in [m] \setminus \{s, \tilde{s}\}$ must satisfy

$$r(\mathcal{S} \cup \{s\} \; ; \; \boldsymbol{\theta}(t)) - r(\mathcal{S} \cup \{\tilde{s}\} \; ; \; \boldsymbol{\theta}(t)) \; \geq \; C \; . \tag{11}$$

Based on $C$-separability, the number of tests required for identifying $\mathcal{S}^\star(\boldsymbol{\theta}(t))$ will then be inversely proportional to $C^2$ (Zhou et al., 2014). However, it is impossible to ensure $C$-separability for the function $r(\cdot \; ; \; \boldsymbol{\theta}(t))$ at round $t \in \mathbb{N}$, even when the reward function $r(\cdot \; ; \; \boldsymbol{\mu})$ at the true mean $\boldsymbol{\mu}$ is $C$-separable. We empirically show that the cumulative number of non-separable instances increases with time $t$. Figure 1, for any $t$, shows the number of times the reward function evaluated at $s \leq t$ is non-separable. Here, by "non-separable", we mean that the reward difference is smaller than $C$, i.e., $r(\mathcal{S} \cup \{s\} \; ; \; \boldsymbol{\theta}(t)) - r(\mathcal{S} \cup \{\tilde{s}\} \; ; \; \boldsymbol{\theta}(t)) \leq C$, where $C$ is the minimum separability at the true mean. Furthermore, in Figure 1 we plot the function $\frac{1}{\Delta^2_{\min}(\boldsymbol{\mu})} \log(t)$ and observe that the cumulative number of non-separable instances grows faster than $\frac{1}{\Delta^2_{\min}(\boldsymbol{\mu})} \log(t)$, which is not desirable, as it can result in sub-optimal regret.

**Quantization.** To circumvent the non-separability of the reward function at the posterior means, we use *quantized rewards* as the test scores for GTO. Specifically, we use a uniform quantizer to discretize the reward values. This quantizer splits the interval $[0, M]$ into equal sub-intervals of width $\Delta/2B$, where the quantization level $\Delta$ will be specified in Section 4 to ensure sublinear regret[2]. Based on this, we split the reward range into $L = \lceil 2BM/\Delta \rceil$ intervals, where each interval $k \in [L-1]$ is defined as

$$I_j := \left( \frac{(i-1)\Delta}{2B}, \frac{i\Delta}{2B} \right] \; , \quad k < L \; , \tag{12}$$

$$I_K := \left( \frac{(L-1)\Delta}{2B}, M \right] \; . \tag{13}$$

---

[2]If $2BM/\Delta$ is not an integer, we absorb the missing fraction in the last interval, making it shorter than the preceding ones.

---

**Algorithm 2:** Oracle($\boldsymbol{\theta}$)

---

**Input:** Parameter $\boldsymbol{\theta}$, quatization level $\Delta$, cardinality $K$, parameter $p$

**1** *Initialize* # tests $\ell = O(\frac{1}{\tilde{q}^2(t)\Delta^2} \log m)$, Test matrix $\mathbf{A} \in \{0,1\}^{\ell \times m}$ such that $A_{i,j} \sim \mathsf{Bern}(p)$

**2** **for** $j = 1 \dots \ell$ **do**

**3**    Evaluate the average reward function $r(\mathbf{A}_j, \boldsymbol{\theta})$ at the input $\boldsymbol{\theta}$

**4**    Assign the quantized score $\xi(r(\mathbf{A}_j \; ; \; \boldsymbol{\theta}))$ to the test $j$ according to (14)

**5** **end**

**6** Evaluate the grading function using the decoding matrix $\mathbf{A}$ according to (10)

**Output:** $\mathcal{S}(t)$ : arms having the top-$K$ grades

---

Furthermore, we denote the set of quantization levels by $\mathcal{L} := \{\Delta/2B, \cdots, M\}$. Accordingly, for any $\boldsymbol{\theta} \in [0,1]^m$ and $\mathcal{S} \in \mathcal{I}$, the uniform quantizer $\xi : [0,B] \mapsto \mathcal{L}$ is specified by

$$\xi(r(\mathcal{S} \; ; \; \boldsymbol{\theta})) := \underset{\ell \in \mathcal{L}}{\arg\min} \; |r(\mathcal{S} \; ; \; \boldsymbol{\theta}) - \ell| \; . \tag{14}$$

**Decoding.** Note that quantization alone does not guarantee the separability of the reward defined in (11) evaluated at *every* test. The reason is that the reward evaluations for the sets $\mathcal{S} \cup \{s\}$ and $\mathcal{S} \cup \{\tilde{s}\}$ may be mapped to the same quantization level, even though we have $r(\mathcal{S} \cup \{s\} \; ; \; \boldsymbol{\theta}(t)) > r(\mathcal{S} \cup \{\tilde{s}\} \; ; \; \boldsymbol{\theta}(t))$ by Assumption 6. Accordingly, at each round $t$, let us denote the set of *unique* (or non-repeated) test scores by $\mathcal{I}_{\mathsf{nr}}(t)$. Specifically, for any pair of distinct tests $\mathcal{S}, \mathcal{S}' \in \mathcal{I}_{\mathsf{nr}}(t)$ such that $|\mathcal{S}| = |\mathcal{S}'|$, it satisfies that $\xi(r(\mathcal{S} \; ; \; \boldsymbol{\theta}(t))) \neq \xi(r(\mathcal{S}' \; ; \; \boldsymbol{\theta}(t)))$. In the decoding step, we leverage the fact that our quantization scheme enables us to sufficiently distinguish the tests contained in $\mathcal{I}_{\mathsf{nr}}(t)$.

**Arm selection.** Let us denote the subset of arms obtained under a test matrix $\mathbf{A}$ and the scoring function $\rho$ by $\mathrm{GTO}(\mathbf{A}, \rho)$. At each round $t$, the GT+QTS algorithm uses the quantized average reward function $\xi(r(\mathbf{A}_i, \boldsymbol{\theta}(t)))$ as the scoring function for each test $i \in [\ell]$. Subsequently, the set of arms to be chosen at time $t$ is set to $\mathcal{S}(t) := \mathrm{GTO}(\mathbf{A}, \xi(r(\cdot, \boldsymbol{\theta}(t-1))))$. The entire algorithm is presented in Algorithm 1.

## 4    Main Results: Efficiency and Regret

In this section, we present the performance guarantees of the proposed GT+QTS algorithm. Specifically, we investigate two key performance metrics of the algorithm: (1) the efficiency of the GTO measured in terms of the number of reward evaluations required in each step, and (2) the average cumulative regret incurred by the GT+QTS algorithm. We show that the GT+QTS algorithm achieves the same order-wise regret guarantee as the combinatorial Thompson sampling using an exact oracle (Wang & Chen, 2018; Perrault et al., 2021), while exponentially reducing the number of reward function evaluations. We begin with the results on the efficiency of the GTO.

**Efficiency.** A naïve approach to finding the optimal super-arm in each round is to evaluate the functional value at every subset in $\mathcal{I}$ at the current estimate of $\boldsymbol{\theta}(t)$ at time $t$. However, this approach requires an exponential number of reward evaluations. An exact oracle may not require an exponential number of evaluations, owing to the separability in Assumption 6. We will first describe a baseline approach, called Oracle$_+$, that provides an *exact* solution leveraging Assumption 6, with the reward function evaluations scaling linearly with respect to the number of base arms. Subsequently, we will demonstrate that the GTO described in Section 3 identifies the optimal arm with high probability, requiring only $O(\log m)$ function evaluations, thereby exponentially reducing the complexity compared to the baseline approach.

**Oracle$_+$:** The baseline approach is a direct consequence of Assumption 6. Since the separability assumption is valid for *any* subset $\mathcal{S}$, for any parameter $\boldsymbol{\theta} \in [0,1]^m$ we may set $\mathcal{S} = \emptyset$. By this choice, for any $s \in \mathcal{S}^\star(\boldsymbol{\theta})$ and $\tilde{s} \notin \mathcal{S}^\star(\boldsymbol{\theta})$, Assumption 6 implies that

$$r(s \; ; \; \boldsymbol{\theta}) > r(\tilde{s} \; ; \; \boldsymbol{\theta}) \; . \tag{15}$$

Oracle$_+$ makes $m$ reward evaluations, each test comprising of a single base arm. It then selects the top $K$ arms with the largest reward values. As a consequence of (15), we immediately conclude that this set of base arms selected by Oracle$_+$ is indeed the optimal super-arm $\mathcal{S}^\star(\boldsymbol{\theta})$. Therefore, Oracle$_+$ requires $m$ (linear) reward function evaluations. Next, we analyze the number of tests required by the GTO.

**GTO:** For any separable function, the number of tests required by the GTO is of the order $O(\log m)$, where the constants depend on the quantization level $\Delta$, the set $\mathcal{I}_{\mathsf{nr}}(t)$, as well as the test matrix parameter $p$. For characterizing the number of tests required by GTO, let us denote the probability that any test $\mathcal{S}$ belongs to the set of non-repeated tests at time $t$ by

$$q(t) := \mathbb{P}(\mathcal{S} \in \mathcal{I}_{\mathsf{nr}}(t)) . \tag{16}$$

The following lemma formalizes the number of tests the GTO requires to compute the optimal super-arm at each time $t \in \mathbb{N}$.

**Lemma 1.** *For any $\delta \in (0,1)$,*

$$\ell = \frac{8M^2 B^2}{\Delta^2 p^4 (1-p)^2 q^2(t)} \log\left(\frac{K(m-K)}{\delta}\right) \tag{17}$$

*tests are sufficient for the GTO to identify an optimal super-arm in each round with probability at least $1 - \delta$.*

From (17), we observe that the GTO requires $O(\log m)$ tests to identify the optimal super-arm in each round with probability at least $1 - \delta$. Hence, in the regime of a large number of base arms, the GTO *significantly* reduces the number of reward evaluations required to find the optimal super-arm in each round. We also observe that the number of tests depends on $q(t)$, which is unknown. This can be resolved by adopting an estimator for estimating $q(t)$ based on the group tests. Let us define

$$\hat{q}(t) := \frac{1}{\ell} \sum_{j \in [\ell]} \mathbb{1}\{\mathbf{A}_j \in \mathcal{I}_{\mathsf{nr}}(t)\} . \tag{18}$$

We show that using $O(\frac{1}{\varepsilon^2} \log \frac{1}{\delta})$ samples, we have an $\varepsilon-$accurate estimate of $q(t)$ with a high probability, i.e., $\mathbb{P}(|\hat{q}(t) - q(t)| > \varepsilon) \leq \delta$. Both $\Delta$ and $q(t)$ capture the granularity of the reward function in identifying the optimal super-arm. We will show that $\Delta$ is chosen based on the CDF $\mathbb{F}_{\boldsymbol{\mu}}$ of the arm gaps $\Delta_{\min}(\boldsymbol{\mu})$, which captures the gap between reward due to optimal arm and any other arm. The smaller the quantization level $\Delta$, the more tests are required to find the optimal super-arm. Similarly, if we face a reward function in which many tests get mapped to the same score, it is unlikely that we save much by leveraging group testing.

**Regret analysis.** Next, we characterize the regret of the GT+QTS algorithm. As a first step, we establish that our quantization scheme for super-arm selection does not compromise the achievable regret. In other words, the regret achieved after reward quantization is equivalent to the regret with unquantized rewards. For this, we introduce a few notations. Let $\mathcal{T}(\boldsymbol{\theta})$ denote the *set* of *all* optimal super-arms with respect to the parameter $\boldsymbol{\theta}$, i.e., $\mathcal{S}^\star(\boldsymbol{\theta}) \in \mathcal{T}(\boldsymbol{\theta})$. Next, corresponding to the function $\xi$ specified in (14) we define

$$\mathcal{T}_\xi(\boldsymbol{\theta}) := \arg\max_{\mathcal{S} \in \mathcal{I}} \xi(r(\mathcal{S} ; \boldsymbol{\theta})) . \tag{19}$$

We show that, with a high probability, the set of optimal super-arms with respect to the quantized reward $\mathcal{T}_\xi(\boldsymbol{\mu})$ is contained in the set of optimal super-arms with respect to the true reward $\mathcal{T}(\boldsymbol{\mu})$. This is necessary to achieve sublinear regret, as we aim to converge to one of the optimal super-arms after quantization.

**Lemma 2.** *For any $\gamma \in [0, 1/2]$, let us set $\Delta := \mathbb{F}_{\boldsymbol{\mu}}^{-1}(\gamma)$. For the quantization scheme described in Section 3, with probability at least $1 - 2\gamma$ we have $\mathcal{T}_\xi(\boldsymbol{\mu}) \subseteq \mathcal{T}(\boldsymbol{\mu})$.*

Next, we provide an upper bound on the average cumulative regret that GT+QTS can achieve. We note that this is similar to the regret bound reported when access to an exact oracle for the CTS algorithm is available.

**Theorem 1** (Achievable regret). *Under Assumptions 1–6, by setting $\delta = \frac{1}{t^2}$, and the quantization level $\Delta := \mathbb{F}_{\boldsymbol{\mu}}^{-1}(\gamma)$ for any $\gamma \in [0, 1/2]$, with probability at least $1 - 2\gamma$, the average cumulative regret of the GT+QTS algorithm, conditioned on the bandit instance $\boldsymbol{\mu}$, satisfies*

$$\mathfrak{R}(T) \;\leq\; \sum_{i \in [m]} (2 \log K + 6) \, B^2 \times \frac{\log(2^m |\mathcal{I}| T)}{\min\limits_{\mathcal{S}:i \in \mathcal{S}} \left( \Delta(\mathcal{S}, \boldsymbol{\mu}) - \frac{\Delta_{\min}(\boldsymbol{\mu})}{2} - (K^2 + 2) B \varepsilon \right)}$$
$$+ \left( 13\alpha \frac{8}{\varepsilon^2} \left( \frac{4}{\varepsilon^2} + 1 \right)^K \log \frac{K}{\varepsilon^2} + \frac{\pi^2}{6} + m \left( \frac{K^2}{\varepsilon^2} + 1 \right) \right) \Delta_{\max}(\boldsymbol{\mu}) \;, \tag{20}$$

*where $\alpha \in \mathbb{R}_+$ is a constant, and $\varepsilon \in \mathbb{R}_+$ is chosen as*

$$\varepsilon \;<\; \frac{\Delta_{\min}(\boldsymbol{\mu})}{4B(K^2 + 2)} \;. \tag{21}$$

A probabilistic guarantee on the *average* cumulative regret in Theorem 1 might seem counterintuitive. However, note that the randomness (and hence the probabilistic guarantee) arises from the distribution $\mathbb{F}_{\boldsymbol{\mu}}$ on the minimum gap $\Delta_{\min}(\boldsymbol{\mu})$, and the average regret is conditioned on the bandit instance $\boldsymbol{\mu}$. The regret bound in Theorem 1 matches that of the CTS algorithm with an exact oracle (Wang & Chen, 2018) order-wise, i.e., both have the same regret bound of $O(m\Delta_{\min}^{-1}(\boldsymbol{\mu}) \log K \log T)$, despite GT+QTS requiring exponentially fewer reward function evaluations. We note that the regret remains linear in $m$. The numerator of the first term in the summand, i.e., $\log(2^m|\mathcal{I}|T)$, can be decomposed into two parts. The first part is $m \log(2|\mathcal{I}|)$, and the second part is $\log T$. The first part depends on $T$ through $\log T$, but it is **independent of** $m$, and the second part is **independent of** $T$ but depends on $m$ linearly. Since the second part is independent of $T$, it does not contribute to the regret, and therefore, the regret will be specified only by the first term. In other words, after summing both terms $m$ times, we get the total regret of $m \log T + m^2$, which is $O(m \log T)$. Comparing the bound in Theorem 1 to that of Wang & Chen (2018, Theorem 1), we observe that GTO only adds a constant term of $\frac{\pi^2}{6} \Delta_{\max}(\boldsymbol{\mu})$ to the regret bound.

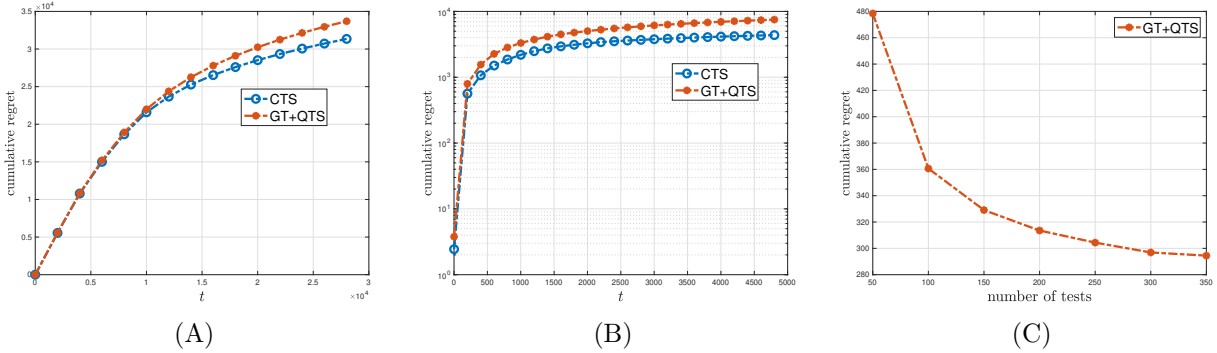

(A)  (B)  (C)

Figure 2: Average cumulative regret versus time $t$ for (A) a linear reward function and (B) a non-linear 2-layer ANN function, respectively. (C) Average cumulative regret versus number of tests $\ell$.

*Proof.* We provide an overview of the key steps and defer the details of the proof to Appendix D. From the definition of regret in (6), with probability at least $1 - 2\gamma$ we have

$$\mathfrak{R}(T) \;=\; \sum_{t=1}^{T} \mathbb{E}[\mathbb{1}\{\mathcal{S}(t) \notin \mathcal{T}(\boldsymbol{\mu})\} \times \Delta(\mathcal{S}(t), \boldsymbol{\mu})]$$
$$\leq\; \sum_{t=1}^{T} \mathbb{E}[\mathbb{1}\{\mathcal{S}(t) \notin \mathcal{T}_\xi(\boldsymbol{\mu})\} \times \Delta(\mathcal{S}(t), \boldsymbol{\mu})] \;, \tag{22}$$

where (22) is a result of Lemma 2. Next, we decompose the upper bound on the regret in (22) based on three events. The first event captures the instances at which we select a sub-optimal super-arm due to inaccurate

sample mean estimates. The second event considers time instances at which the sample mean is close to the true mean, yet we select a sub-optimal super-arm since our posterior mean has a considerable deviation from the true mean. Finally, the third event considers the instances where the posterior mean of the *selected* super-arm is close to the true mean, and yet, we select a sub-optimal super-arm. The key challenge arises in upper-bounding the regret due to this third event. Specifically, the proof for this term in Wang & Chen (2018) relies on assuming access to an *exact* oracle – an assumption that we have dispensed with. We show that GTO is sufficient to guarantee constant regret for the third set of events. □

## 5  Experiments

In this section, we provide empirical results to assess the performance of GT+QTS and compare it against the state-of-the-art CTS algorithm provided in (Wang & Chen, 2018) equipped with Oracle$_+$ described in Section 4 as the exact oracle. We consider two reward functions: linear rewards and non-linear rewards modeled as the output of a 2-layer artificial neural network. We conduct experiments on both synthetic data and real-world datasets.

**Hyperparameters for GT-QTS.** For GT-QTS, we have two hyperparameters: (1) the entries of the encoding and decoding matrix $\mathbf{A}$, and (2) the number of tests $\ell$. For (1), the entries $A_{i,j}$ are drawn from a Bernoulli distribution with parameter $p$. In Lemma 1, we show that choosing $p = 0.5$ results in an $O(\log m)$ number of tests. In the experiments, we observe that choosing each column $\mathbf{A}_{:,j} \sim \text{Bern}(\boldsymbol{\theta}(t))$ results in a better empirical performance. This implies that the probability of choosing arm $i \in [m]$ in a test is directly proportional to the current posterior average $\theta_i(t)$ of the arm. We adopt this choice in all the experiments. Finally, a sufficient number of tests $\ell$ has been enumerated in Lemma 1. Furthermore, we provide an ablation study of how the number of tests impacts regret in Figure 2(C).

**Linear rewards.** In this experiment, for any set $\mathcal{S} \in \mathcal{I}$ and any $\boldsymbol{\theta} \in [0,1]^m$, we define the reward function as $r(\mathcal{S} \,;\, \boldsymbol{\theta}) = \sum_{i \in \mathcal{S}} \theta_i$. We set $m = 5000$ arms, and the mean vector $\boldsymbol{\mu}$ is sampled uniformly randomly from $[0,1]^{5000}$. Furthermore, the cardinality constraint is set to $K = 5$ arms. For this experiment, we choose $\boldsymbol{\mu}$ such that $\Delta_{\min}(\boldsymbol{\mu})$ is at least 0.25, and we set $\Delta = 0.25$. Consequently, the group testing oracle requires approximately 302 reward evaluations (order-wise), versus the exact oracle, which requires 5000 reward evaluations in each iteration. Hence, the baseline method (CTS) requires $16\times$ more reward evaluations compared to GT+QTS. However, the regret due to CTS and GT+QTS is comparable, and GT+QTS has a slightly larger regret compared to CTS, as observed in Fig. 2(A).

Furthermore, we empirically observe that the number of tests prescribed by theory is excessive, and in practice, much fewer tests are sufficient to guarantee similar cumulative regret. To showcase this, we vary the number of group tests in the GT+QTS algorithm and plot the average cumulative regret against the number of group tests in Figure 2(C) computed at $T = 10,000$. Figure 2(C) confirms that as few as $\ell = 200$ tests are sufficient for the regret to be within 5% of the regret at the prescribed number of $\ell \approx 310$ tests.

**Two-layer neural network rewards.** Next, we evaluate the performance of GT+QTS on nonlinear mean reward functions. Specifically, we choose a 2-layer NN with 20 neurons and a sigmoid activation function. For any set $\mathcal{S}$ and parameter $\boldsymbol{\theta}$, the mean reward is $r(\mathcal{S} \,;\, \boldsymbol{\theta}) = \langle \mathbf{w}_2, \sigma(\mathbf{W}_1 \boldsymbol{\theta}_{\mathcal{S}}) \rangle$, where $\sigma(\cdot)$ denotes the sigmoid activation. The weights are uniformly sampled from a normal distribution, and then we take the absolute value to make all weights positive. We choose $m = 1000$ arms, which are uniformly sampled at random from $[0,1]^{1000}$. Furthermore, the weight matrices are sampled such that we have $\Delta_{\min}(\boldsymbol{\mu}) \geq 0.2$, and we set $\Delta = 0.2$. Figure 2(B) illustrates the cumulative regret of the CTS algorithm and the GT+QTS algorithm, which follow the same order in $T$.

**Impact of $\Delta$.** To assess the impact of the quantization interval $\Delta$ on the regret, we perform an ablation study for varying levels of $\Delta$ and its impact on the average cumulative regret. Specifically, we adopt the linear reward setting, i.e., $r(\mathcal{S}, \boldsymbol{\theta}) = \sum_{i \in \mathcal{S}} \theta_i$, and we set $m = 505$ arms, of which we select a subset of $K = 5$ arms in each iteration. We have performed 50 independent trials, and the average results of this experiment are provided in Figure 3(A). We observe that as the quantization level increases, the regret increases. This is expected since the larger the gap, the less distinguishable the sub-optimal super-arms

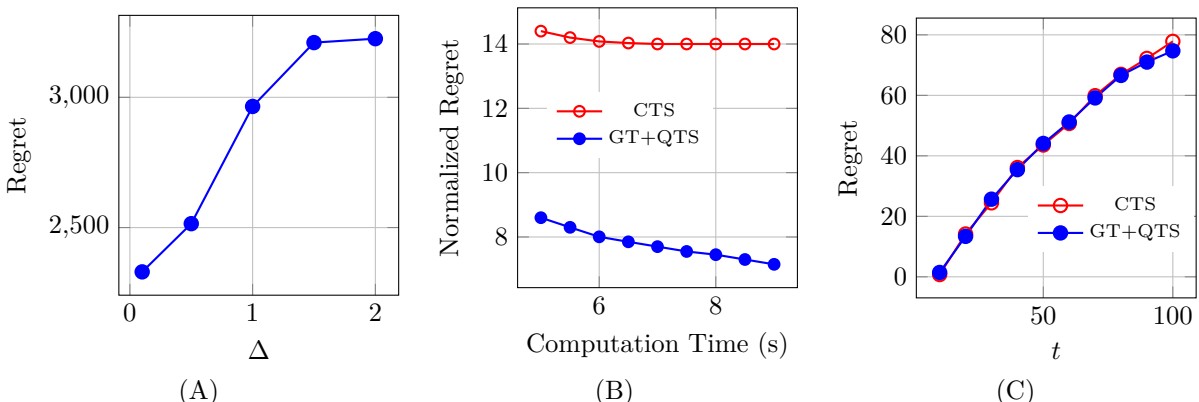

Figure 3: (A) Average cumulative regret versus $\Delta$ for GT-QTS, when $T = 5000$. (B) Normalized average regret versus computation time. (C) Average cumulative regret versus time for `MovieLens-100K`.

are from the optimal ones, resulting in a higher probability of erroneously selecting sub-optimal super-arms. Selecting such super-arms inevitably leads to increased regret.

**Computation Times.** In Figure 3(B), we plot the regret of the CTS and GT-QTS algorithms for various values of computation time. We adopt a 2-layer ANN with 50 neurons as the reward function for this experiment, where we set $m = 500$ arms, and we select $K = 5$ arms in each round. The experiment is averaged over 100 independent trials, and the simulation is performed using Python 3.9.21 on a MacBook Pro with an M1 Pro processor and 16 GB of RAM. Figure 3(B) shows that GT-QTS significantly outperforms CTS in terms of regret for specified values of computation time.

**Real-world dataset.** To capture the regret efficiency of the proposed GT-QTS algorithm, we also test its performance on the `MovieLens-100K` dataset (Harper & Konstan, 2015), which consists of $100,000$ ratings from 943 users for 1682 movies. Each user is asked to annotate a minimum of 20 movies. In the experiment, we uniformly randomly select a user and adopt the goal of recommending a set of 5 movies that match the user's preferences. Here, movies are designed as arms of a multi-armed bandit. In each round, the learner selects a super-arm of size $K = 5$ and receives feedback (rating) for each arm. The feedback is a Bernoulli random variable with mean value set to the (scaled) original rating from the dataset. Subsequently, the reward corresponding to the super-arm is chosen as the sum of feedback from the selected arms. For implementing the GT-QTS algorithm, we set $\Delta = 0.2$, which readily follows from the observation that movie ratings lie in the set $\{1, 2, 3, 4, 5\}$. In Figure 3(C), we compare the performance of GT-QTS against CTS, which shows a comparable regret performance to the baseline, which assumes access to an exact oracle.

## Conclusions

We investigate the problem of regret minimization in combinatorial semi-bandits. Existing approaches assume the existence of an exact oracle, which may not always be computationally viable. To circumvent this issue, we establish a novel connection between group testing and combinatorial bandits. We propose a new arm-selection strategy that combines a group testing oracle with a Thompson sampling-based super-arm selection strategy. Under a probabilistic assumption on the minimum separation over the class of bandit instances, the proposed GT-QTS algorithm has two key advantages: 1) it is significantly more efficient compared to the exact oracle since it requires exponentially fewer reward evaluations at each step, and 2) it preserves the regret guarantee of the state-of-the-art method order-wise. We provide numerical evaluations to bolster our analytical claims. A promising direction is to investigate the impact of using group testing for CMABs in dynamic environments, i.e, when the ground truth model $\boldsymbol{\mu}$ is expected to evolve over time, and as a result we face a *sequence* of models $\{\mu(t) : t \in [T]\}$. We conjecture that an extension of the GT-QTS algorithm to dynamic environments is highly non-trivial, owing to challenges induced by the estimator (which now has to track an evolving model), and the group testing-based super-arm selection, which crucially hinges on designing an appropriate $\Delta$ – a quantity which is also evolving in the dynamic setting.

**Acknowledgments**

This work was supported by the Rensselaer-IBM AI Research Collaboration (http://airc.rpi.edu), part of the IBM AI Horizons Network (http://ibm.biz/AIHorizons).

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

## A    Practical Implications of the GT-based CMAB Framework

In a broad range of applications, the decision space's cardinality is vast. Furthermore, performing an experiment for each possible decision can have extensive time or monetary costs. For instance, in precision medicine, treatments have to be tailored to individual patients based on genetic, clinical, and lifestyle factors. The number of such treatment regimens, especially when combined with the combination of medications, their dosage, and schedules, results in thousands to millions of candidate treatments (i.e., arms). Furthermore, in many such applications, performing each experiment has a high monetary cost and/or is time-consuming.

- **Experimental time and cost in genomics:** In functional genomics, researchers aim to understand how combinations of gene perturbations (e.g., knockouts or activations) influence a biological phenotype—such as drug response or cell viability. A common objective is identifying the subset of genes whose knock-out maximally affects a target phenotype. This naturally lends itself to a combinatorial bandit formulation, where arms correspond to genes, and each super-arm is a subset of genes to be perturbed. Evaluating a super-arm involves a costly biological experiment – often requiring weeks or months of lab work and substantial monetary investment. Thus, the utility function is expensive to evaluate, and the goal is to minimize regret while limiting the number of such high-cost experiments.

- **Monetary cost per query:** Performing clinical trials in the above genomics example can also be highly costly. In another example, consider the problem of crafting adversarial examples against commercial image classification APIs (e.g., Google Vision or Clarifai). Each query to the API incurs a monetary cost. This setting can be framed as an *active learning* problem, where the learner selects a batch of queries, receives labels from the API, and iteratively searches for adversarial inputs – while minimizing financial expenditure. Here, the oracle is the API, and each super-arm (batch) query is associated with a dollar cost.

## B    Proof of Lemma 1

At any instant $t \in \mathbb{N}$, choose an optimal arm $s \in \mathcal{S}^\star(\boldsymbol{\theta}(t))$ and a sub-optimal arm $\tilde{s} \notin \mathcal{S}^\star(\boldsymbol{\theta}(t))$. For accurate prediction, the arm grade $\phi_s(t)$ for arm $s$ should be more than $\phi_{\tilde{s}}(t)$ assigned to arm $\tilde{s}$. Let us denote the $i^{\text{th}}$ column of any matrix $\mathbf{A}$ by $\mathbf{A}_{:,i}$. Finding the difference between the arm grades, we have

$$\phi_s(t) - \phi_{\tilde{s}}(t) \;=\; \langle \mathbf{A}_{:,s} \,,\, \rho(t) \rangle - \langle \mathbf{A}_{:,\tilde{s}} \,,\, \boldsymbol{\rho}(t) \rangle \tag{23}$$

$$= \sum_{j=1}^{\ell} \underbrace{(\mathbf{A}_{j,s} - \mathbf{A}_{j,\tilde{s}})\, \rho_j(t)}_{:=Z_j(t)} \;. \tag{24}$$

Furthermore, we have

$$\mathbb{E}[Z_j(t)] \;= \mathbb{E}\left[(\mathbf{A}_{j,s} - \mathbf{A}_{j,\tilde{s}})\rho_j(t)\right] \tag{25}$$

$$= \; \mathbb{E}\left[(\mathbf{A}_{j,s} - \mathbf{A}_{j,\tilde{s}}) \times \xi(r(\mathbf{A}_j \,;\, \boldsymbol{\theta}(t)))\right] \tag{26}$$

$$= \sum_{\mathcal{S} \subseteq [m]} \left(\mathbb{1}\{s \in \mathcal{S}\} - \mathbb{1}\{\tilde{s} \in \mathcal{S}\}\right) \times \xi(r(\mathcal{S} \,;\, \boldsymbol{\theta}(t))) \times \mathbb{P}(\mathcal{S} \in \mathbf{A}) \tag{27}$$

$$= \sum_{\mathcal{S} \subseteq [m]: s \in \mathcal{S}, \tilde{s} \in \mathcal{S}} \left( \underbrace{\mathbb{1}\{s \in \mathcal{S}\} - \mathbb{1}\{\tilde{s} \in \mathcal{S}\}}_{=0} \right) \times \xi(r(\mathcal{S} \,;\, \boldsymbol{\theta}(t))) \times \mathbb{P}(\mathcal{S} \in \mathbf{A})$$

$$+ \sum_{\mathcal{S} \subseteq [m]: s \notin \mathcal{S}, \tilde{s} \notin \mathcal{S}} \left( \underbrace{\mathbb{1}\{s \in \mathcal{S}\} - \mathbb{1}\{\tilde{s} \in \mathcal{S}\}}_{=0} \right) \times \xi(r(\mathcal{S} \,;\, \boldsymbol{\theta}(t))) \times \mathbb{P}(\mathcal{S} \in \mathbf{A})$$

$$+ \sum_{\mathcal{S} \subseteq [m]: s \in \mathcal{S}, \tilde{s} \notin \mathcal{S}} \left( \underbrace{\mathbb{1}\{s \in \mathcal{S}\} - \mathbb{1}\{\tilde{s} \in \mathcal{S}\}}_{=1} \right) \times \xi(r(\mathcal{S} \,;\, \boldsymbol{\theta}(t))) \times \mathbb{P}(\mathcal{S} \in \mathbf{A})$$

$$+ \sum_{\mathcal{S} \subseteq [m]:s \notin \mathcal{S}, \tilde{s} \in \mathcal{S}} \left( \underbrace{\mathbb{1}\{s \in \mathcal{S}\} - \mathbb{1}\{\tilde{s} \in \mathcal{S}\}}_{=-1} \right) \times \xi(r(\mathcal{S} \; ; \; \boldsymbol{\theta}(t))) \times \mathbb{P}(\mathcal{S} \in \mathbf{A}) \tag{28}$$

$$= \sum_{\mathcal{S} \subseteq [m] \setminus \{s, \tilde{s}\}} Q\Big( r(\mathcal{S} \cup \{s\} \; ; \; \boldsymbol{\theta}(t)) \Big) \times \mathbb{P}(\mathcal{S} \cup \{s\} \in \mathbf{A})$$

$$- \sum_{\mathcal{S} \subseteq [m] \setminus \{s, \tilde{s}\}} Q\Big( r(\mathcal{S} \cup \{\tilde{s}\} \; ; \; \boldsymbol{\theta}(t)) \Big) \times \mathbb{P}(\mathcal{S} \cup \{\tilde{s}\} \in \mathbf{A}) \tag{29}$$

$$= p(1-p) \sum_{\mathcal{S} \subseteq [m] \setminus \{s, \tilde{s}\}} \Big( Q\Big( r(\mathcal{S} \cup \{s\} \; ; \; \boldsymbol{\theta}(t)) \Big) - Q\Big( r(\mathcal{S} \cup \{\tilde{s}\} \; ; \; \boldsymbol{\theta}(t)) \Big) \Big)$$

$$\times p^{|\mathcal{S}|}(1-p)^{m-|\mathcal{S}|-2} \; . \tag{30}$$

Next, let us recall the definitions of the set of repeated tests $\mathcal{I}_{\mathsf{nr}}(t)$. Accordingly, we have

$$\mathbb{E}[Z_j(t)] = p(1-p) \sum_{\mathcal{S} \subseteq [m] \setminus \{s, \tilde{s}\}} \Big( Q\Big( r(\mathcal{S} \cup \{s\} \; ; \; \boldsymbol{\theta}(t)) \Big) - Q\Big( r(\mathcal{S} \cup \{\tilde{s}\} \; ; \; \boldsymbol{\theta}(t)) \Big) \Big)$$

$$\times p^{|\mathcal{S}|}(1-p)^{m-|\mathcal{S}|-2} \tag{31}$$

$$= p(1-p) \sum_{\mathcal{S} \subseteq [m] \setminus \{s, \tilde{s}\}: \mathcal{S} \cup \{s\} \in \mathcal{I}_{\mathsf{nr}}(t)} \underbrace{\Big( Q\Big( r(\mathcal{S} \cup \{s\} \; ; \; \boldsymbol{\theta}(t)) \Big) - Q\Big( r(\mathcal{S} \cup \{\tilde{s}\} \; ; \; \boldsymbol{\theta}(t)) \Big) \Big)}_{\geq \frac{\Delta_{\min}(\boldsymbol{\mu})}{2B}}$$

$$\times p^{|\mathcal{S}|}(1-p)^{m-|\mathcal{S}|-2}$$

$$+ p(1-p) \sum_{\mathcal{S} \subseteq [m] \setminus \{s, \tilde{s}\}: \mathcal{S} \cup \{s\} \notin \mathcal{I}_{\mathsf{nr}}(t)} \underbrace{\Big( Q\Big( r(\mathcal{S} \cup \{s\} \; ; \; \boldsymbol{\theta}(t)) \Big) - Q\Big( r(\mathcal{S} \cup \{\tilde{s}\} \; ; \; \boldsymbol{\theta}(t)) \Big) \Big)}_{\geq 0}$$

$$\times p^{|\mathcal{S}|}(1-p)^{m-|\mathcal{S}|-2} \tag{32}$$

$$\geq \frac{\Delta}{2B} p(1-p) \sum_{\mathcal{S} \subseteq [m] \setminus \{s, \tilde{s}\}: \mathcal{S} \cup \{s\} \in \mathcal{I}_{\mathsf{nr}}(t)} p^{|\mathcal{S}|}(1-p)^{m-|\mathcal{S}|-2} \tag{33}$$

$$= \frac{\Delta}{2B} p(1-p) \mathbb{P}\Big( \{\mathcal{S} \in \mathcal{I}_{\mathsf{nr}}(t) : s \in \mathcal{S}\} \Big) \tag{34}$$

$$= \frac{\Delta}{2B} p^2 (1-p) q(t) \; , \tag{35}$$

where (35) follows from the fact that $\mathbb{P}(\mathcal{S} \in \mathcal{I}_{\mathsf{nr}}(t), s \in \mathcal{S}) = \mathbb{P}(\mathcal{S} \in \mathcal{I}_{\mathsf{nr}}(t) \mid s \in \mathcal{S}) \mathbb{P}(s \in \mathcal{S}) = pq(t)$. Furthermore, since we have $Z_j(t) \in [-M, M]$ as per Assumption 4, by Hoeffding's inequality we have

$$\mathbb{P}\Big( \phi_s(t) - \phi_{\tilde{s}}(t) \leq 0 \Big) \leq \exp\left( -\frac{\ell \Delta^2 p^4 (1-p)^2 q^2(t)}{8 M^2 B^2} \right) \; . \tag{36}$$

Finally, noting that there are $K(m-K)$ possible ways to choose $s$ and $\tilde{s}$, taking a union bound along with (36) concludes the proof.

**Estimating** $q$: First, note that $\hat{q}(t)$ is an unbiased estimator of $q(t)$. This is because

$$\mathbb{E}[\hat{q}(t)] = \frac{1}{\ell} \mathbb{E}\left[ \sum_{j \in \ell} \mathbb{1}\{\mathbf{A}_j \in \mathcal{I}_{\mathsf{nr}}(t)\} \right] \tag{37}$$

$$= \frac{1}{\ell} \sum_{j \in \ell} \mathbb{P}\Big( \mathbf{A}_j \in \mathcal{I}_{\mathsf{nr}}(t) \Big) \tag{38}$$

$$= \mathbb{P}\Big( \mathcal{I}_{\mathsf{nr}}(t) \Big) \tag{39}$$

$$= q(t) \; . \tag{40}$$

Furthermore, since $\hat{q}(t)$ is an unbiased estimator of $q$, using the Hoeffding's inequality, we obtain that for any $\varepsilon \in \mathbb{R}_+$ and $\delta \in (0,1)$,

$$\ell = \frac{1}{2\varepsilon^2} \log \frac{1}{\delta} \tag{41}$$

tests are sufficient to ensure that

$$\mathbb{P}\Big(|\hat{q}(t) - q| > \varepsilon\Big) \leq \delta . \tag{42}$$

## C   Proof of Lemma 2

First, we will show that with a high probability, we have $\mathcal{T}_Q(\boldsymbol{\mu}) \cap \mathcal{T}(\boldsymbol{\mu}) \neq \emptyset$. Note that

$$\mathbb{P}\Big(\mathcal{T}_Q(\boldsymbol{\mu}) \cap \mathcal{T}(\boldsymbol{\mu}) = \emptyset\Big)$$

$$= \mathbb{P}\Big(\exists\, \mathcal{S} \in \mathcal{T}(\boldsymbol{\mu}), \exists\, \mathcal{S}' \in \mathcal{T}_Q(\boldsymbol{\mu}) : \mathcal{S} \notin \mathcal{T}_Q(\boldsymbol{\mu}) \text{ and } \mathcal{S}' \notin \mathcal{T}(\boldsymbol{\mu})\Big) \tag{43}$$

$$\leq \mathbb{P}\Big(\exists\, \mathcal{S} \in \mathcal{T}(\boldsymbol{\mu}), \exists\, \mathcal{S}' \in \mathcal{T}_Q(\boldsymbol{\mu}) : r(\mathcal{S}\,;\,\boldsymbol{\mu}) - r(\mathcal{S}'\,;\,\boldsymbol{\mu}) \geq \Delta_{\min}(\boldsymbol{\mu})\Big) \tag{44}$$

$$= \mathbb{P}\Big(\exists\, \mathcal{S} \in \mathcal{T}(\boldsymbol{\mu}), \exists\, \mathcal{S}' \in \mathcal{T}_Q(\boldsymbol{\mu}) :$$
$$\qquad r(\mathcal{S}\,;\,\boldsymbol{\mu}) - \xi(r(\mathcal{S}'\,;\,\boldsymbol{\mu})) + \xi(r(\mathcal{S}'\,;\,\boldsymbol{\mu})) - r(\mathcal{S}'\,;\,\boldsymbol{\mu}) \geq \Delta_{\min}(\boldsymbol{\mu})\Big) \tag{45}$$

$$\leq \mathbb{P}\Big(\exists\, \mathcal{S} \in \mathcal{T}(\boldsymbol{\mu}), \exists\, \mathcal{S}' \in \mathcal{T}_Q(\boldsymbol{\mu}) :$$
$$\qquad r(\mathcal{S}\,;\,\boldsymbol{\mu}) - \xi(r(\mathcal{S}\,;\,\boldsymbol{\mu})) + \xi(r(\mathcal{S}'\,;\,\boldsymbol{\mu})) - r(\mathcal{S}'\,;\,\boldsymbol{\mu}) \geq \Delta_{\min}(\boldsymbol{\mu})\Big) \tag{46}$$

$$\leq \mathbb{P}\Big(\exists\, \mathcal{S} \in \mathcal{T}(\boldsymbol{\mu}), \exists\, \mathcal{S}' \in \mathcal{T}_Q(\boldsymbol{\mu}) : \frac{\Delta}{4B} + \frac{\Delta}{4B} \geq \Delta_{\min}(\boldsymbol{\mu})\Big) \tag{47}$$

$$\leq \mathbb{P}\Big(\frac{\Delta}{2B} \geq \Delta_{\min}(\boldsymbol{\mu})\Big) \tag{48}$$

$$= \mathbb{P}\Big(\frac{\mathbb{F}_{\boldsymbol{\mu}}^{-1}(\gamma)}{2B} \geq \Delta_{\min}(\boldsymbol{\mu})\Big) \tag{49}$$

$$\leq \mathbb{P}\Big(\mathbb{F}_{\boldsymbol{\mu}}^{-1}(\gamma) \geq \Delta_{\min}(\boldsymbol{\mu})\Big) \tag{50}$$

$$= \gamma , \tag{51}$$

where (46) follows from the definition of the set $\mathcal{S}'$, (47) follows from the quantization scheme in (14), (49) holds since we have set $\Delta = \mathbb{F}_{\boldsymbol{\mu}}^{-1}(\gamma)$, and (50) holds since $B$ is the Lipschitz constant in Assumption 2, and it can always be set to be larger than $1/2$, if any $B < 1/2$ satisfies Assumption 2. This proves that $\mathcal{T}_\xi(\boldsymbol{\mu}) \cap \mathcal{T}(\boldsymbol{\mu}) \neq \emptyset$ with probability at least $1 - \gamma$. Next, following a similar line of arguments, we will show that $\mathcal{T}_Q(\boldsymbol{\mu}) \subseteq \mathcal{T}(\boldsymbol{\mu})$ with a high probability. Let us define the event

$$\mathcal{E}(\boldsymbol{\mu}) := \Big\{\mathcal{T}_Q(\boldsymbol{\mu}) \cap \mathcal{T}(\boldsymbol{\mu}) = \emptyset\Big\} . \tag{52}$$

We have,

$$\mathbb{P}\Big(\mathcal{T}(\boldsymbol{\mu}) \subset \mathcal{T}_\xi(\boldsymbol{\mu})\Big)$$

$$= \mathbb{P}\Big(\exists\, \mathcal{S}' \in \mathcal{T}_\xi(\boldsymbol{\mu}) : \mathcal{S}' \notin \mathcal{T}(\boldsymbol{\mu})\Big) \tag{53}$$

$$= \mathbb{P}\Big(\exists\, \mathcal{S}' \in \mathcal{T}_\xi(\boldsymbol{\mu}) : \mathcal{S}' \notin \mathcal{T}(\boldsymbol{\mu}) \mid \mathcal{E}(\boldsymbol{\mu})\Big)\mathbb{P}\Big(\mathcal{E}(\boldsymbol{\mu})\Big) + \mathbb{P}\Big(\exists\, \mathcal{S}' \in \mathcal{T}_\xi(\boldsymbol{\mu}) : \mathcal{S}' \notin \mathcal{T}(\boldsymbol{\mu}) \mid \overline{\mathcal{E}(\boldsymbol{\mu})}\Big)\mathbb{P}\Big(\overline{\mathcal{E}(\boldsymbol{\mu})}\Big) \tag{54}$$

$$\overset{(51)}{<} \mathbb{P}\Big(\exists\, \mathcal{S}' \in \mathcal{T}_\xi(\boldsymbol{\mu}) : \mathcal{S}' \notin \mathcal{T}(\boldsymbol{\mu}) \mid \overline{\mathcal{E}(\boldsymbol{\mu})}\Big) + \gamma \tag{55}$$

$$= \mathbb{P}\Big(\exists\, \mathcal{S}' \in \mathcal{T}_\xi(\boldsymbol{\mu}), \tilde{\mathcal{S}} \in \mathcal{T}_\xi(\boldsymbol{\mu}) \cap \mathcal{T}(\boldsymbol{\mu}) : r(\tilde{\mathcal{S}} \,;\, \boldsymbol{\mu}) - r(\mathcal{S}' \,;\, \boldsymbol{\mu}) \geq \Delta_{\min}(\boldsymbol{\mu}) \mid \overline{\mathcal{E}(\boldsymbol{\mu})}\Big) + \gamma \tag{56}$$

$$= \mathbb{P}\Big(\exists\, \mathcal{S}' \in \mathcal{T}_\xi(\boldsymbol{\mu}), \tilde{\mathcal{S}} \in \mathcal{T}_\xi(\boldsymbol{\mu}) \cap \mathcal{T}(\boldsymbol{\mu}) : r(\tilde{\mathcal{S}} \,;\, \boldsymbol{\mu}) - \xi(r(\mathcal{S}' \,;\, \boldsymbol{\mu}))$$
$$+ \xi(r(\mathcal{S}' \,;\, \boldsymbol{\mu})) - r(\mathcal{S}' \,;\, \boldsymbol{\mu}) \geq \Delta_{\min}(\boldsymbol{\mu}) \mid \overline{\mathcal{E}(\boldsymbol{\mu})}\Big) + \gamma \tag{57}$$

$$= \mathbb{P}\Big(\exists\, \mathcal{S}' \in \mathcal{T}_\xi(\boldsymbol{\mu}), \tilde{\mathcal{S}} \in \mathcal{T}_\xi(\boldsymbol{\mu}) \cap \mathcal{T}(\boldsymbol{\mu}) : r(\tilde{\mathcal{S}} \,;\, \boldsymbol{\mu}) - \xi(r(\tilde{\mathcal{S}} \,;\, \boldsymbol{\mu}))$$
$$+ \xi(r(\mathcal{S}' \,;\, \boldsymbol{\mu})) - r(\mathcal{S}' \,;\, \boldsymbol{\mu}) \geq \Delta_{\min}(\boldsymbol{\mu}) \mid \overline{\mathcal{E}(\boldsymbol{\mu})}\Big) + \gamma \tag{58}$$

$$\leq \mathbb{P}\Big(\exists\, \mathcal{S}' \in \mathcal{T}_\xi(\boldsymbol{\mu}), \tilde{\mathcal{S}} \in \mathcal{T}_\xi(\boldsymbol{\mu}) \cap \mathcal{T}(\boldsymbol{\mu}) : \frac{\Delta}{4B} + \frac{\Delta}{4B} > \Delta_{\min}(\boldsymbol{\mu}) \mid \overline{\mathcal{E}(\boldsymbol{\mu})}\Big) + \gamma \tag{59}$$

$$\leq \mathbb{P}\Big(\exists\, \mathcal{S}' \in \mathcal{T}_\xi(\boldsymbol{\mu}), \tilde{\mathcal{S}} \in \mathcal{T}_\xi(\boldsymbol{\mu}) \cap \mathcal{T}(\boldsymbol{\mu}) : \frac{1}{2}\Delta > \Delta_{\min}(\boldsymbol{\mu}) \mid \overline{\mathcal{E}(\boldsymbol{\mu})}\Big) + \gamma \tag{60}$$

$$\leq \mathbb{P}\Big(\frac{1}{2}\Delta > \Delta_{\min}(\boldsymbol{\mu})\Big) \tag{61}$$

$$\leq 2\gamma \,, \tag{62}$$

where (61) follows from the fact that the events $\overline{\mathcal{E}(\boldsymbol{\mu})}$ and $\{\frac{1}{2}\Delta_{\min}(\boldsymbol{\mu})\}$ are independent of each other, since the distribution of $\Delta_{\min}(\boldsymbol{\mu})$ is a property of the environment, and does not depend on the event $\mathcal{E}(\boldsymbol{\mu})$. This concludes our proof.

## D    Proof of Theorem 1

Similarly to (Wang & Chen, 2018), we begin by defining a few events that are instrumental in characterizing the upper bound on the average regret. First, let us denote the number of times that any arm $i \in [m]$ is sampled until time $t \in \mathbb{N}$ by $T_i(t)$. Furthermore, let us denote the sample mean for any arm $i \in [m]$ at time $t \in \mathbb{N}$ by $\bar{\mu}_i(t)$. Accordingly, let us define

1. $\mathcal{A}(t) := \{\mathcal{S}(t) \notin \mathcal{T}_\xi(\boldsymbol{\mu})\}$.

2. $\mathcal{B}(t) := \left\{\exists i \in \mathcal{S}(t) : |\bar{\mu}_i(t) - \mu_i| > \frac{\varepsilon}{|\mathcal{S}(t)|}\right\}$.

3. $\mathcal{C}(t) := \left\{||\boldsymbol{\theta}_{\mathcal{S}(t)}(t) - \boldsymbol{\mu}_{\mathcal{S}(t)}||_1 > \frac{\Delta(\mathcal{S}(t), \boldsymbol{\mu})}{B} - \frac{\Delta_{\min}(\boldsymbol{\mu})}{2B} - (K^2 + 2)\varepsilon\right\}$.

With a probability at least $1 - 2\gamma$, we can decompose the regret as follows.

$$\mathfrak{R}(T) = \sum_{t=1}^{T} \mathbb{E}\Big[\mathbb{1}\{\mathcal{S}(t) \notin \mathcal{T}(\boldsymbol{\mu})\} \times \Delta(\mathcal{S}(t), \boldsymbol{\mu})\Big] \tag{63}$$

$$\leq \sum_{t=1}^{T} \mathbb{E}\Big[\mathbb{1}\{\mathcal{A}(t))\} \times \Delta(\mathcal{S}(t), \boldsymbol{\mu})\Big] \tag{64}$$

$$\leq \underbrace{\sum_{t=1}^{T} \mathbb{E}\Big[\mathbb{1}\{\mathcal{A}(t) \cap \mathcal{B}(t)\} \times \Delta(\mathcal{S}(t), \boldsymbol{\mu})\Big]}_{A_1} + \underbrace{\sum_{t=1}^{T} \mathbb{E}\Big[\mathbb{1}\{\mathcal{A}(t) \cap \overline{\mathcal{B}(t)} \cap \mathcal{C}(t)\} \times \Delta(\mathcal{S}(t), \boldsymbol{\mu})\Big]}_{A_2}$$

$$+ \underbrace{\sum_{t=1}^{T} \mathbb{E}\Big[\mathbb{1}\{\mathcal{A}(t) \cap \overline{\mathcal{C}(t)}\} \times \Delta(\mathcal{S}(t), \boldsymbol{\mu})\Big]}_{A_3}, \tag{65}$$

where (64) is a result of Lemma 2. Next, we find an upper bound for each of the terms $A_1$, $A_2$ and $A_3$ to recover the regret bound in Theorem 1.

**Upper-bounding $A_1$:** First, we leverage (Wang & Chen, 2018, Lemma 1) to find an upper bound on $A_1$, which we state below for completeness.

**Lemma 3** ( Wang & Chen (2018)). *In Algorithm 1, we have*

$$\mathbb{E}\left[\sum_{t=1}^{T}\mathbb{1}\{i \in \mathcal{S}(t), \ |\bar{\mu}_i(t) - \mu_i| > \varepsilon\}\right] \leq 1 + \frac{1}{\varepsilon^2} \ . \tag{66}$$

Leveraging Lemma 3, it can be readily verified that the regret due to $A_1$ can be upper bounded as

$$A_1 \leq \left(\frac{mK^2}{\varepsilon^2} + m\right)\Delta_{\max}(\boldsymbol{\mu}) \ . \tag{67}$$

**Upper-bounding $A_2$:** Next, we provide an upper-bound for the term $A_2$. First, note that under the event $\overline{\mathcal{B}(t)} \cap \mathcal{C}(t)$, the event

$$\mathcal{G}(t) := \left\{||\boldsymbol{\theta}_{\mathcal{S}(t)}(t) - \bar{\boldsymbol{\mu}}_{\mathcal{S}(t)}||_1 > \frac{\Delta(\mathcal{S}(t), \boldsymbol{\mu})}{B} - \frac{\Delta_{\min}(\boldsymbol{\mu})}{2B} - (K^2 + 1)\varepsilon\right\} \tag{68}$$

holds. Furthermore, let us define the event

$$\mathcal{H}(t) := \left\{\sum_{i \in \mathcal{S}(t)} \frac{1}{T_i(t)} \leq \frac{2\left(\frac{\Delta(\mathcal{S}(t), \boldsymbol{\mu})}{B} - \frac{\Delta_{\min}(\boldsymbol{\mu})}{2B} - (K^2 + 2)\varepsilon\right)^2}{\log(2^m|\mathcal{I}|T)}\right\} \ . \tag{69}$$

Subsequently, we may expand the event $\mathcal{G}(t)$ as

$$\mathcal{G}(t) = \mathcal{G}(t) \cap \mathcal{H}(t) \ \cup \ \mathcal{G}(t) \cap \overline{\mathcal{H}(t)} \ . \tag{70}$$

Next, note that using (Perrault et al., 2021, Lemma 2), it can be readily verified that

$$\mathbb{P}\left(\mathcal{G}(t) \cap \mathcal{H}(t)\right) \leq \frac{1}{T} \quad \forall t \in \mathbb{N} \ . \tag{71}$$

Hence, what remains is to upper-bound the term

$$\sum_{t=1}^{T}\mathbb{E}\left[\mathbb{1}\{\mathcal{G}(t) \cap \overline{\mathcal{H}(t)}\} \times \Delta(\mathcal{S}(t), \boldsymbol{\mu})\right] \ . \tag{72}$$

Similarly to Wang & Chen (2018), under the event $\overline{\mathcal{H}(t)}$, we define a function that upper-bounds the regret at time $t$ due to the super-arm $\mathcal{S}(t)$. Specifically, for any arm $i \in \mathcal{S}(t)$, let $g_i(T_i(t))$ denote this function, and we show that $\sum_{i \in \mathcal{S}(t)} g_i(T_i(t)) \geq \Delta(\mathcal{S}(t), \boldsymbol{\mu})$. Finally, we have

$$\sum_{t=1}^{T}\mathbb{E}\left[\mathbb{1}\{\mathcal{G}(t) \cap \overline{\mathcal{H}(t)}\} \times \Delta(\mathcal{S}(t), \boldsymbol{\mu})\right] \leq \sum_{t \in \mathbb{N}}\sum_{i \in \mathcal{S}(t)} g_i(T_i(t)) \ . \tag{73}$$

For any $i \in [m]$, let us define the function

$$g_i(n) := \begin{cases} \Delta_{\max}(\boldsymbol{\mu}), & \text{if } n = 0 \\[2mm] 2B\sqrt{\dfrac{\log(2^m|\mathcal{I}|T)}{n}}, & \text{if } 1 \leq n \leq L_{i,1} \\[4mm] \dfrac{2B\log(2^m|\mathcal{I}|T)}{n\min\limits_{\mathcal{S}:i\in\mathcal{S}}\left(\frac{\Delta(\mathcal{S},\boldsymbol{\mu})}{B} - \frac{\Delta_{\min}(\boldsymbol{\mu})}{2} - (K^2 + 2)\varepsilon\right)}, & \text{if } L_{i,1} < n \leq L_{i,2} \\[4mm] 0, & \text{if } n > L_{i,2} \end{cases} \ , \tag{74}$$

where we have defined

$$L_{i,1} := \frac{\log(2^m|\mathcal{I}|T)}{\min\limits_{\mathcal{S}:i\in\mathcal{S}}\left(\frac{\Delta(\mathcal{S},\boldsymbol{\mu})}{B} - \frac{\Delta_{\min}(\boldsymbol{\mu})}{2B} - (K^2+2)\varepsilon\right)^2} \;, \tag{75}$$

and,

$$L_{i,2} := \frac{K\log(2^m|\mathcal{I}|T)}{\min\limits_{\mathcal{S}:i\in\mathcal{S}}\left(\frac{\Delta(\mathcal{S},\boldsymbol{\mu})}{B} - \frac{\Delta_{\min}(\boldsymbol{\mu})}{2B} - (K^2+2)\varepsilon\right)^2} \;. \tag{76}$$

Next, we verify that the function $g_i$ defined in (74) satisfies the condition that $\sum_{i\in\mathcal{S}(t)} g_i(T_i(t)) \geq \Delta(\mathcal{S}(t),\boldsymbol{\mu})$ for every $t\in\mathbb{N}$. First, note that if there exists arms $j\in\mathcal{S}(t)$ such that $T_j(t) = 0$, we have

$$\sum_{i\in\mathcal{S}(t)} g_i(T_i(t)) \;\geq\; g_j(T_j(t)) \tag{77}$$

$$= \Delta_{\max}(\boldsymbol{\mu}) \tag{78}$$

$$\geq \Delta(\mathcal{S}(t),\boldsymbol{\mu}) \;. \tag{79}$$

Next, if there exists an arm $j\in\mathcal{S}(t)$ such that

$$1 \leq T_j(t) \leq \frac{\log(2^m|\mathcal{I}|T)}{\left(\frac{\Delta(\mathcal{S}(t),\boldsymbol{\mu})}{B} - \frac{\Delta_{\min}(\boldsymbol{\mu})}{2B} - (K^2+2)\varepsilon\right)^2} \;, \tag{80}$$

which implies that $1 \leq T_j(t) \leq L_{i,1}$, we have

$$\sum_{i\in\mathcal{S}(t)} g_i(T_i(t)) \;\geq\; g_j(T_j(t)) \tag{81}$$

$$= 2\sqrt{\frac{\log(2^m|\mathcal{I}|T)}{T_j(t)}} \tag{82}$$

$$\geq 2\left(\frac{\Delta(\mathcal{S}(t),\boldsymbol{\mu})}{B} - \frac{\Delta_{\min}(\boldsymbol{\mu})}{2B} - (K^2+2)\varepsilon\right) \tag{83}$$

$$\geq \Delta(\mathcal{S}(t),\boldsymbol{\mu}) \;, \tag{84}$$

where (84) holds since we have chosen $\varepsilon\in\mathbb{R}_+$ such that

$$\varepsilon \;<\; \frac{\Delta_{\min}(\boldsymbol{\mu})}{4B(K^2+2)} \;. \tag{85}$$

Next, if, for all $i\in\mathcal{S}(t)$ we have

$$T_i(t) > \frac{\log(2^m|\mathcal{I}|T)}{\left(\frac{\Delta(\mathcal{S}(t),\boldsymbol{\mu})}{B} - \frac{\Delta_{\min}(\boldsymbol{\mu})}{2B} - (K^2+2)\varepsilon\right)^2} \;, \tag{86}$$

we can decompose $\mathcal{S}(t)$ into three disjoint subsets. Specifically, we define

$$\mathcal{S}_1(t) := \{i\in\mathcal{S}(t) : T_i(t) \leq L_{i,1}\} \;, \tag{87}$$
$$\mathcal{S}_2(t) := \{i\in\mathcal{S}(t) : L_{i,1} < T_i(t) \leq L_{i,2}\} \;, \tag{88}$$
$$\mathcal{S}_3(t) := \{i\in\mathcal{S}(t) : T_i(t) > L_{i,2}\} \;. \tag{89}$$

Subsequently, we have

$$\sum_{i\in\mathcal{S}(t)} g_i(T_i(t))$$

$$= \sum_{i \in \mathcal{S}_1(t)} g_i(T_i(t)) + \sum_{i \in \mathcal{S}_2(t)} g_i(T_i(t)) \tag{90}$$

$$= \sum_{i \in \mathcal{S}_1(t)} 2B \sqrt{\frac{\log(2^m |\mathcal{I}|T)}{T_i(t)}} + \sum_{i \in \mathcal{S}_2(t)} \frac{2B \log(2^m |\mathcal{I}|T)}{T_i(t) \min\limits_{\mathcal{S}:i \in \mathcal{S}} \left( \frac{\Delta(\mathcal{S},\boldsymbol{\mu})}{B} - \frac{\Delta_{\min}(\boldsymbol{\mu})}{2B} - (K^2+2)\varepsilon \right)} \tag{91}$$

$$\geq \sum_{i \in \mathcal{S}_1(t)} 2B \sqrt{\frac{\log(2^m |\mathcal{I}|T)}{T_i(t)}} + \sum_{i \in \mathcal{S}_2(t)} \frac{2B \log(2^m |\mathcal{I}|T)}{T_i(t) \left( \frac{\Delta(\mathcal{S}(t),\boldsymbol{\mu})}{B} - \frac{\Delta_{\min}(\boldsymbol{\mu})}{2B} - (K^2+2)\varepsilon \right)} \tag{92}$$

$$= \sum_{i \in \mathcal{S}_1(t)} \frac{2B \log(2^m |\mathcal{I}|T)}{T_i(t) \left( \frac{\Delta(\mathcal{S}(t),\boldsymbol{\mu})}{B} - \frac{\Delta_{\min}(\boldsymbol{\mu})}{2B} - (K^2+2)\varepsilon \right)} \times \sqrt{\frac{T_i(t) \left( \frac{\Delta(\mathcal{S}(t),\boldsymbol{\mu})}{B} - \frac{\Delta_{\min}(\boldsymbol{\mu})}{2B} - (K^2+2)\varepsilon \right)^2}{\log(2^m |\mathcal{I}|T)}}$$
$$+ \sum_{i \in \mathcal{S}_2(t)} \frac{2B \log(2^m |\mathcal{I}|T)}{T_i(t) \left( \frac{\Delta(\mathcal{S}(t),\boldsymbol{\mu})}{B} - \frac{\Delta_{\min}(\boldsymbol{\mu})}{2B} - (K^2+2)\varepsilon \right)} \tag{93}$$

$$\geq \sum_{i \in \mathcal{S}_1(t)} \frac{2B \log(2^m |\mathcal{I}|T)}{T_i(t) \left( \frac{\Delta(\mathcal{S}(t),\boldsymbol{\mu})}{B} - \frac{\Delta_{\min}(\boldsymbol{\mu})}{2B} - (K^2+2)\varepsilon \right)} + \sum_{i \in \mathcal{S}_2(t)} \frac{2B \log(2^m |\mathcal{I}|T)}{T_i(t) \left( \frac{\Delta(\mathcal{S}(t),\boldsymbol{\mu})}{B} - \frac{\Delta_{\min}(\boldsymbol{\mu})}{2B} - (K^2+2)\varepsilon \right)} \tag{94}$$

$$= \frac{2B \log(2^m |\mathcal{I}|T)}{\left( \frac{\Delta(\mathcal{S}(t),\boldsymbol{\mu})}{B} - \frac{\Delta_{\min}(\boldsymbol{\mu})}{2B} - (K^2+2)\varepsilon \right)} \times \left( \sum_{i \in \mathcal{S}(t)} \frac{1}{T_i(t)} - \sum_{i \in \mathcal{S}_3(t)} \frac{1}{T_i(t)} \right) \tag{95}$$

$$\geq \frac{2B \log(2^m |\mathcal{I}|T)}{\left( \frac{\Delta(\mathcal{S}(t),\boldsymbol{\mu})}{B} - \frac{\Delta_{\min}(\boldsymbol{\mu})}{2B} - (K^2+2)\varepsilon \right)}$$
$$\times \left( \frac{2 \left( \frac{\Delta(\mathcal{S}(t),\boldsymbol{\mu})}{B} - \frac{\Delta_{\min}(\boldsymbol{\mu})}{2B} - (K^2+2)\varepsilon \right)^2}{\log(2^m |\mathcal{I}|T)} - \frac{\left( \frac{\Delta(\mathcal{S}(t),\boldsymbol{\mu})}{B} - \frac{\Delta_{\min}(\boldsymbol{\mu})}{2B} - (K^2+2)\varepsilon \right)^2}{\log(2^m |\mathcal{I}|T)} \right) \tag{96}$$

$$\tag{97}$$

$$\geq \Delta(\mathcal{S}(t), \boldsymbol{\mu}) \,, \tag{98}$$

where (94) uses the fact that

$$T_i(t) > \frac{\log(2^m |\mathcal{I}|T)}{\left( \frac{\Delta(\mathcal{S}(t),\boldsymbol{\mu})}{B} - \frac{\Delta_{\min}(\boldsymbol{\mu})}{2B} - (K^2+2)\varepsilon \right)^2} \,, \quad \forall \ i \in \mathcal{S}(t) \,, \tag{99}$$

and (96) holds due to the event $\overline{\mathcal{H}(t)}$ along with the definition of the set $\mathcal{S}_3(t)$, and (97) holds by the choice of $\varepsilon \in \mathbb{R}_+$. Finally, if all $i \in \mathcal{S}(t)$ satisfy $T_i(t) > L_{i,2}$, we have

$$\sum_{i \in \mathcal{S}(t)} \frac{1}{T_i(t)} \leq \sum_{i \in \mathcal{S}(t)} \frac{1}{L_{i,2}} \tag{100}$$

$$= \sum_{i \in \mathcal{S}(t)} \frac{\min\limits_{\mathcal{S}:i \in \mathcal{S}} \left( \frac{\Delta(\mathcal{S},\boldsymbol{\mu})}{B} - \frac{\Delta_{\min}(\boldsymbol{\mu})}{2B} - (K^2+2)\varepsilon \right)^2}{K \log(2^m |\mathcal{I}|T)} \tag{101}$$

$$\leq \frac{\left( \frac{\Delta(\mathcal{S}(t),\boldsymbol{\mu})}{B} - \frac{\Delta_{\min}(\boldsymbol{\mu})}{2B} - (K^2+2)\varepsilon \right)^2}{\log(2^m |\mathcal{I}|T)} \,, \tag{102}$$

which is in contradiction with the event $\overline{\mathcal{H}(t)}$. Hence, we have shown that under the event $\overline{\mathcal{H}(t)}$, the functions $g_i$ satisfy the inequality $\sum\limits_{i \in \mathcal{S}(t)} g_i(T_i(t)) \geq \Delta(\mathcal{S}(t), \boldsymbol{\mu})$. Finally, summing up the $g_i(T_i(t))$ functions over time and the set of arms, following a similar procedure to (Wang & Chen, 2018), we obtain that

$$A_2 \leq 2m\Delta_{\max}(\boldsymbol{\mu}) + \sum_{i \in [m]} (2 \log K + 6) \frac{B \log(2^m |\mathcal{I}|T)}{\min\limits_{\mathcal{S}:i \in \mathcal{S}} \left( \frac{\Delta(\mathcal{S},\boldsymbol{\mu})}{B} - \frac{\Delta_{\min}(\boldsymbol{\mu})}{2B} - (K^2+2)\varepsilon \right)} \,. \tag{103}$$

**Upper-bounding $A_3$:** Finally, we turn our attention to upper-bounding $A_3$. Before analyzing the upper bound, let us lay down a few notations and definitions required in the analysis. Let $\boldsymbol{\theta}$, $\bar{\boldsymbol{\theta}} \in [0,1]^m$ and $\mathcal{Z} \subseteq [m]$. Accordingly, let us define $\boldsymbol{\theta}' := (\bar{\boldsymbol{\theta}}_{\mathcal{Z}}, \boldsymbol{\theta}_{\bar{\mathcal{Z}}})$ as a vector, whose $i^{\text{th}}$ coordinate has the same value as the $i^{\text{th}}$ coordinate of $\bar{\boldsymbol{\theta}}$ if $i \in \mathcal{Z}$, and otherwise, it has the same value as the $i^{\text{th}}$ coordinate of $\boldsymbol{\theta}$. Let $\mathcal{S}_Q^\star(\boldsymbol{\mu}) \in \mathcal{T}_\xi(\boldsymbol{\mu})$ denote one of the optimal super-arms with respect to the quantized reward function. Furthermore, for any choice of $\bar{\boldsymbol{\theta}}$ and $\mathcal{Z}$ such that $||\bar{\boldsymbol{\theta}}_{\mathcal{Z}} - \mu_{\mathcal{Z}}||_\infty \leq \varepsilon$, let us consider the following properties of the vector $\boldsymbol{\theta}'$.

P1. $\mathcal{Z} \subseteq \mathcal{S}_Q^\star(\boldsymbol{\theta}')$

P2. Either $\mathcal{S}_Q^\star(\boldsymbol{\theta}') \in \mathcal{T}_\xi(\boldsymbol{\mu})$, or $||\boldsymbol{\theta}'_{\mathcal{S}_Q^\star(\boldsymbol{\theta}')} - \boldsymbol{\mu}_{\mathcal{S}_Q^\star(\boldsymbol{\theta}')}||_1 > \frac{1}{B}\Delta(\mathcal{S}_Q^\star(\boldsymbol{\theta}'), \boldsymbol{\mu}) - \frac{1}{2B}\Delta_{\min}(\boldsymbol{\mu}) - (K^2 + 1)\varepsilon$,

Furthermore, for any $\mathcal{Z} \subseteq [m]$ and $\boldsymbol{\theta}, \bar{\boldsymbol{\theta}} \in [0,1]^m$ satisfying $||\bar{\boldsymbol{\theta}}_{\mathcal{Z}} - \boldsymbol{\mu}_{\mathcal{Z}}||_\infty \leq \varepsilon$, let us define the event

$$\mathcal{E}_{\mathcal{Z},1}(\boldsymbol{\theta}) := \Big\{ \text{properties P1 and P2 hold for } \mathcal{Z} \subseteq [m] \text{ and } \boldsymbol{\theta} \in [0,1]^m \Big\}. \tag{104}$$

Additionally, let us define the event

$$\mathcal{M}(t) := \Big\{ \mathcal{S}(t) \neq \mathcal{S}^\star(\boldsymbol{\theta}(t)) \Big\}. \tag{105}$$

For upper-bounding $A_3$, we decompose the event $\mathcal{A}(t) \cap \bar{\mathcal{C}}(t)$ as follows.

$$\mathcal{A}(t) \cap \overline{\mathcal{C}(t)} = \mathcal{A}(t) \cap \overline{\mathcal{C}(t)} \cap \mathcal{M}(t) \ \cup \ \mathcal{A}(t) \cap \overline{\mathcal{C}(t)} \cap \overline{mcM(t)}. \tag{106}$$

Leveraging Lemma 1, we have that at any time $t \in \mathbb{N}$, $\mathbb{P}(\mathcal{M}(t)) \leq \frac{1}{t^2}$. Hence, we have

$$\sum_{t=1}^{T} \mathbb{E}\left[\mathbb{1}\{\mathcal{M}(t)\} \times \Delta(\mathcal{S}(t), \boldsymbol{\mu})\right] < \Delta_{\max}(\boldsymbol{\mu}) \sum_{t=1}^{\infty} \mathbb{P}(\mathcal{M}(t)) \tag{107}$$

$$= \frac{\pi^2}{6}\Delta_{\max}(\boldsymbol{\mu}). \tag{108}$$

Next, we upper-bound the regret due to $A_3$ under the event $\overline{\mathcal{M}(t)}$, i.e., when the GTO returns the same super-arm as the exact oracle. We emphasize that under the event $\mathcal{A}(t) \cap \overline{\mathcal{C}(t)} \cap \overline{\mathcal{M}(t)}$, the analysis does not reduce to the analysis of the CTS algorithm Wang & Chen (2018), since, an exact oracle operates on the *true* reward function $r(\cdot \ ; \ \cdot)$, whereas the GTO operates on the quantized reward function $\xi(r(\cdot \ ; \ \cdot))$. Next, we prove that if the event $\mathcal{A}(t) \cap \overline{\mathcal{C}(t)} \cap \mathcal{M}(t)$ occurs, then it implies that there exists a set $\mathcal{Z} \subseteq \mathcal{S}_Q^\star(\boldsymbol{\mu})$ such that the event $\mathcal{E}_{\mathcal{Z},1}(\boldsymbol{\theta}(t))$ occurs. Before formally proving this statement, let us understand its implication. If there exists $\mathcal{Z} \subseteq \mathcal{S}_Q^\star(\boldsymbol{\mu})$, $\mathcal{Z} \neq \emptyset$, such that $\mathcal{E}_{\mathcal{Z},1}(\boldsymbol{\theta}(t))$ occurs, then it immediately implies that $||\boldsymbol{\theta}_{\mathcal{Z}}(t) - \boldsymbol{\mu}_{\mathcal{Z}}||_\infty > \varepsilon$. This is because, if $||\boldsymbol{\theta}_{\mathcal{Z}}(t) - \boldsymbol{\mu}_{\mathcal{Z}}||_\infty \leq \varepsilon$, then $\boldsymbol{\theta}(t)$ becomes a candidate choice for $\boldsymbol{\theta}'$, and thus, either 1) $\mathcal{S}(t) \in \mathcal{T}_\xi(\boldsymbol{\mu})$, (hence contradicting the event $\mathcal{A}(t)$) or 2) $||\boldsymbol{\theta}_{\mathcal{S}(t)}(t) - \boldsymbol{\mu}_{\mathcal{S}(t)}||_1 > \frac{1}{B}\Delta(\mathcal{S}(t), \boldsymbol{\mu}) - \frac{1}{2B}\Delta_{\min}(\boldsymbol{\mu}) - (K^2 + 1)\varepsilon$ (hence, contradicting the event $\overline{\mathcal{C}(t)}$). Subsequently, we can leverage (Wang & Chen, 2018, Lemma 3) which provides an upper bound on the number of times that the event $\mathcal{E}_{\mathcal{Z},2}(t) := \{||\boldsymbol{\theta}_{\mathcal{Z}}(t) - \boldsymbol{\mu}_{[}\mathcal{Z}]||_\infty > \varepsilon\}$ occurs.

**Lemma 4** (Wang & Chen (2018)). *We have*

$$\sum_{t=1}^{T} \mathbb{E}\left[\mathbb{1}\{\mathcal{A}(t) \cap \overline{\mathcal{C}(t)} \cap \overline{\mathcal{M}(t)} \cap \mathcal{E}_{\mathcal{Z},2}(t)\}\right] \leq 13\alpha\frac{8}{\varepsilon^2}\left(\frac{4}{\varepsilon^2} + 1\right)^K \log\frac{K}{\varepsilon^2}, \tag{109}$$

*where $\alpha \in \mathbb{R}_+$ is a universal constant.*

Hence, we obtain that the regret due to $A_3$ is upper-bounded by

$$A_3 \leq \left(13\alpha\frac{8}{\varepsilon^2}\left(\frac{4}{\varepsilon^2} + 1\right)^K \log\frac{K}{\varepsilon^2} + \frac{\pi^2}{6}\right)\Delta_{\max}(\boldsymbol{\mu}). \tag{110}$$

What remains is to prove the following lemma.

**Lemma 5.** *If the event $\overline{\mathcal{C}(t)} \cap \mathcal{A}(t) \cap \overline{\mathcal{M}(t)}$ happens, then there exists a subset $\mathcal{Z} \subseteq \mathcal{S}_Q^\star(\boldsymbol{\mu})$, $\mathcal{Z} \neq \emptyset$, such that $\mathcal{E}_{\mathcal{Z},1}(\boldsymbol{\theta}(t))$ holds.*

*Proof.* First, let us set $\mathcal{Z} = \mathcal{S}_Q^\star(\boldsymbol{\mu})$. Accordingly, we define the vector $\boldsymbol{\theta}'$ such that $||\boldsymbol{\theta}'_{\mathcal{S}_Q^\star(\boldsymbol{\mu})} - \boldsymbol{\mu}_{\mathcal{S}_Q^\star(\boldsymbol{\mu})}||_\infty \leq \varepsilon$. We will show that for any $\mathcal{S}'$ such that $\mathcal{S}' \cap \mathcal{S}_Q^\star(\boldsymbol{\mu}) = \emptyset$, $\mathcal{S}_Q^\star(\boldsymbol{\theta}') \neq \mathcal{S}'$. To verify this, note that

$$\xi(r(\mathcal{S}' \,;\, \boldsymbol{\theta}')) \;=\; \xi(r(\mathcal{S}' \,;\, \boldsymbol{\theta}(t))) \tag{111}$$

$$\leq\; \xi(r(\mathcal{S}(t) \,;\, \boldsymbol{\theta}(t))) \tag{112}$$

$$\overset{(14)}{\leq}\; r(\mathcal{S}(t) \,;\, \boldsymbol{\theta}(t)) + \frac{\Delta_{\min}(\boldsymbol{\mu})}{4B} \tag{113}$$

$$\overset{\overline{\mathcal{C}(t)}}{\leq}\; r(\mathcal{S}(t) \,;\, \boldsymbol{\mu}) + \Delta(\mathcal{S}(t), \boldsymbol{\mu}) - B(K^2 + 1)\varepsilon - \frac{\Delta_{\min}(\boldsymbol{\mu})}{4B} \tag{114}$$

$$<\; r(\mathcal{S}_Q^\star(\boldsymbol{\mu}) \,;\, \boldsymbol{\mu}) - BK\varepsilon - \frac{\Delta_{\min}(\boldsymbol{\mu})}{4B} \tag{115}$$

$$\leq\; r(\mathcal{S}_Q^\star(\boldsymbol{\mu}) \,;\, \boldsymbol{\theta}') + BK\varepsilon - BK\varepsilon - \frac{\Delta_{\min}(\boldsymbol{\mu})}{4B} \tag{116}$$

$$\overset{(14)}{\leq}\; \xi(r(\mathcal{S}_Q^\star(\boldsymbol{\mu}) \,;\, \boldsymbol{\theta}')) \,, \tag{117}$$

where (115) is a consequence of Lemma 2 and (116) follows from Assumption 2. Hence, from (117) we conclude that $\mathcal{S}' \neq \mathcal{S}_Q^\star(\boldsymbol{\theta}')$. So, we have two possibilities for $\mathcal{S}_Q^\star(\boldsymbol{\theta}')$.

a) $\mathcal{S}_Q^\star(\boldsymbol{\mu}) \subseteq \mathcal{S}_Q^\star(\boldsymbol{\theta}')$.

b) Let us define $\mathcal{Z}_1 := \mathcal{S}_Q^\star(\boldsymbol{\theta}') \cap \mathcal{S}_Q^\star(\boldsymbol{\mu})$. Then, we have $\mathcal{Z}_1 \neq \emptyset$.

For the case (a), if $\mathcal{S}_Q^\star(\boldsymbol{\theta}') \notin \mathcal{T}(\boldsymbol{\mu})$, we have

$$r(\mathcal{S}_Q^\star(\boldsymbol{\theta}') \,;\, \boldsymbol{\theta}') \;>\; r(\mathcal{S}_Q^\star(\boldsymbol{\theta}') \,;\, \boldsymbol{\mu}) - BK\varepsilon \tag{118}$$

$$\geq r(\mathcal{S}_Q^\star(\boldsymbol{\mu}) \,;\, \boldsymbol{\mu}) + \Delta(\mathcal{S}_Q^\star(\boldsymbol{\theta}'), \boldsymbol{\mu}) - BK\varepsilon \,, \tag{119}$$

which, along with Assumption 2, implies that

$$||\boldsymbol{\theta}'_{\mathcal{S}_Q^\star(\boldsymbol{\theta}')} - \boldsymbol{\mu}_{\mathcal{S}_Q^\star(\boldsymbol{\theta}')}||_1 \;>\; \frac{\Delta(\mathcal{S}_Q^\star(\boldsymbol{\theta}'), \boldsymbol{\mu})}{B} - K\varepsilon \,, \tag{120}$$

which implies that $\mathcal{E}_{\mathcal{Z},1}(\boldsymbol{\theta}(t))$ holds with $\mathcal{Z} = \mathcal{S}_Q^\star(\boldsymbol{\mu})$. Otherwise, for case (b), we follow the same set of arguments as in (Wang & Chen, 2018, Lemma 2), which concludes the proof. $\qquad\square$

Finally, Theorem 1 is obtained by adding the upeer-bounds obtained due to the terms $A_1$, $A_2$ and $A_3$.

# E   Artificial Neural Network (ANN)

In this section, we show that a 2-layer ANN with sigmoid activation satisfies the separability condition in Assumption (6), with some conditions on the weights. Specifically, consider a 2-layer ANN with the hidden layer weight matrix denoted by $\mathbf{W}_1$ and the output weights denoted by the vector $\mathbf{w}_2$, i.e., for any input $\boldsymbol{\theta}_\mathcal{S} \in [0,1]^m$, the output of the neural network is given by

$$r(\mathcal{S} \,;\, \boldsymbol{\theta}) \;:=\; \Big\langle \mathbf{w}_2 \,,\, \sigma\Big(\mathbf{W}_1 \boldsymbol{\theta}_\mathcal{S}\Big)\Big\rangle \,, \tag{121}$$

where $\sigma(x) := \frac{1}{1+e^{-x}}$ denotes the sigmoid activation function. The result is formally defined next.

**Theorem 2.** *Any 2-layer ANN with the hidden layer $\mathbf{W}_1$ and output weights $\mathbf{w}_2$ is separable, i.e., for any $s \in \mathcal{S}^\star(\boldsymbol{\theta})$ and $\tilde{s} \notin \mathcal{S}^\star(\boldsymbol{\theta})$, and for any $\mathcal{S} \subseteq [m] \setminus \{s, \tilde{s}\}$, we have*

$$r(\mathcal{S} \cup \{s\} ; \boldsymbol{\theta}) - r(\mathcal{S} \cup \{\tilde{s}\} ; \boldsymbol{\theta}) > 0 , \tag{122}$$

*for any $\boldsymbol{\theta} \in [0,1]^m$.*

*Proof.* Let us denote the number of neurons in the hidden layer by $N$. The difference in rewards for any $\boldsymbol{\theta} \in [0,1]^m$ and sets $\mathcal{S}, \mathcal{S}' \subseteq [0,1]^m \setminus \{s, \tilde{s}\}$ can be expanded as

$$r(\mathcal{S} \cup \{s\} ; \boldsymbol{\theta}) - r(\mathcal{S} \cup \{\tilde{s}\} ; \boldsymbol{\theta})$$

$$= \left\langle \mathbf{w}_2 , \underbrace{\sigma\Big(\mathbf{W}_1, \boldsymbol{\theta}_{\mathcal{S} \cup \{s\}}\Big) - \sigma\Big(\mathbf{W}_1, \boldsymbol{\theta}_{\mathcal{S} \cup \{\tilde{s}\}}\Big)}_{:=\mathbf{y}} \right\rangle \tag{123}$$

$$= \sum_{n=1}^{N} w_{2,n} y_n , \tag{124}$$

where $w_{2,n}$ and $y_n$ denote the $n^{\text{th}}$ coordinates of the vector $\mathbf{w}_2$ and $\mathbf{y}$ for any $n \in [N]$. Furthermore, for any set $\mathcal{S} \subseteq [m]$, the $n^{\text{th}}$ coordinate of the vector $\mathbf{v} := \sigma(\mathbf{W}_1 \boldsymbol{\theta}_{\mathcal{S}})$ is given by

$$v_n = \frac{\exp\Big(-\sum_{i \in \mathcal{S}} \theta_i \mathbf{W}_{1,n,i}\Big)}{1 + \exp\Big(-\sum_{i \in \mathcal{S}} \theta_i \mathbf{W}_{1,n,i}\Big)} . \tag{125}$$

Accordingly, we have for any $n \in [N]$,

$$y_n = \frac{\Big(\exp\Big(-\sum_{i \in \mathcal{S}} \theta_i \mathbf{W}_{1,n,i}\Big)\Big)\Big(\exp(-\theta_{\tilde{s}} \mathbf{W}_{1.n,\tilde{s}} - \exp(-\theta_s \mathbf{W}_{1,n,s}))\Big)}{\Big(1 + \exp\Big(-\sum_{i \in \mathcal{S} \cup \{s\}} \theta_i \mathbf{W}_{1,n,i}\Big)\Big)\Big(1 + \exp\Big(-\sum_{i \in \mathcal{S} \cup \{\tilde{s}\}} \theta_i \mathbf{W}_{1,n,i}\Big)\Big)} . \tag{126}$$

Next, let us define the quantities

$$\delta(s, \tilde{s}, n) := \Big(\exp(-\theta_{\tilde{s}} \mathbf{W}_{1.n,\tilde{s}} - \exp(-\theta_s \mathbf{W}_{1,n,s}))\Big) , \tag{127}$$

and for any set $\mathcal{S} \subseteq [m] \setminus \{s, \tilde{s}\}$,

$$\alpha(\mathcal{S} ; n) := \frac{\Big(\exp\Big(-\sum_{i \in \mathcal{S}} \theta_i \mathbf{W}_{1,n,i}\Big)\Big)}{\Big(1 + \exp\Big(-\sum_{i \in \mathcal{S} \cup \{s\}} \theta_i \mathbf{W}_{1,n,i}\Big)\Big)\Big(1 + \exp\Big(-\sum_{i \in \mathcal{S} \cup \{\tilde{s}\}} \theta_i \mathbf{W}_{1,n,i}\Big)\Big)} . \tag{128}$$

Leveraging (127) and (128), we can write (124) as

$$r(\mathcal{S} \cup \{s\} ; \boldsymbol{\theta}) - r(\mathcal{S} \cup \{\tilde{s}\} ; \boldsymbol{\theta}) = \sum_{n=1}^{N} w_{2,n} \times \alpha(\mathcal{S} ; n) \times \delta(s, \tilde{s}, n) . \tag{129}$$

Next, let us set $\mathcal{S} = \mathcal{S}' := \mathcal{S}^\star(\boldsymbol{\theta}) \setminus \{s\}$. Accordingly, we have that

$$r(\mathcal{S}^\star(\boldsymbol{\theta}) ; \boldsymbol{\theta}) - r(\mathcal{S}' \cup \{\tilde{s}\} ; \boldsymbol{\theta}) = \sum_{n=1}^{N} w_{2,n} \times \alpha(\mathcal{S}' ; n) \times \delta(s, \tilde{s}, n) \tag{130}$$

$$\geq \ \Delta_{\min}(\boldsymbol{\theta}) \ . \tag{131}$$

Furthermore, for any set $\mathcal{S} \subseteq [m] \setminus \{s, \tilde{s}\}$ we have

$$r(\mathcal{S} \cup \{s\} \ ; \ \boldsymbol{\theta}) - r(\mathcal{S} \cup \{\tilde{s}\} \ ; \ \boldsymbol{\theta}) \ = \ \sum_{n=1}^{N} w_{2,n} \times \alpha(\mathcal{S} \ ; \ n) \times \delta(s, \tilde{s}, n) \tag{132}$$

$$= \ \sum_{n=1}^{N} w_{2,n} \times \delta(s, \tilde{s}, n) \times \alpha(\mathcal{S}' \ ; \ n) \times \frac{\alpha(\mathcal{S} \ ; \ n)}{\alpha(\mathcal{S}' \ ; \ n)} \ . \tag{133}$$

Defining $\beta := \min_{n \in [N]} \min_{\mathcal{S} \subseteq [m] \setminus \{s, \tilde{s}\}} \frac{\alpha(\mathcal{S} \ ; \ n)}{\alpha(\mathcal{S}' \ ; \ n)}$, we note that $\beta \in \mathbb{R}_+$, since $\alpha(\mathcal{S} \ ; \ n) \in \mathbb{R}_+$ for every $\mathcal{S} \in [m] \setminus \{s, \tilde{s}\}$ and $n \in [N]$. Hence, (133) can be lower bounded as

$$r(\mathcal{S} \cup \{s\} \ ; \ \boldsymbol{\theta}) - r(\mathcal{S} \cup \{\tilde{s}\} \ ; \ \boldsymbol{\theta}) \ \geq \ \beta \sum_{n=1}^{N} w_{2,n} \times \alpha(\mathcal{S}' \ ; \ n) \times \delta(s, \tilde{s}, n) \tag{134}$$

$$\overset{(131)}{\geq} \ \beta \Delta_{\min}(\boldsymbol{\theta}) \tag{135}$$

$$> \ 0 \ . \tag{136}$$

$\square$

