# OpenReview forum: "Combinatorial Multi-armed Bandits: Arm Selection via Group Testing"
_TMLR — Accepted by TMLR_

### Review · Reviewer_EQq6 · 2025-02-16

**Summary Of Contributions:**

The paper introduces a novel approach to the Combinatorial Multi-Armed Bandit (CMAB) problem by integrating group testing (GT) with "quantized Thompson sampling (QTS)" for super-arm selection. The claimed motivation is to replace the exact oracle used in existing methods with a more computationally efficient alternative. The authors propose the GT+QTS algorithm, which reduces the computational complexity of the oracle while maintaining a near-optimal regret. The theoretical analysis is supported by numerical experiments comparing GT+QTS to existing methods.

**Audience:**

Yes

**Claims And Evidence:**

Yes

**Requested Changes:**

1. Expand on the effect of quantization on arm selection errors (theoretically and/or empirically)
2. Expand on the real-world implications of the group testing framework for CMABs
3. Discuss (conjectures are fine, no need to add results) on your approach would perform in dynamic environments
4. Expand on how to tune the hyperparameters you introduce.

**Strengths And Weaknesses:**

Strengths

1. The paper presents a novel combination of group testing with Thompson sampling, which to my knowledge has not been explored in the CMAB context.
2. The proposed GT+QTS algorithm provides an exponential reduction in oracle complexity while achieving a regret comparable to state-of-the-art methods.
3. The authors introduce a "quantized reward function" to ensure the separability assumption, which is a neat trick, with some caveats (see below)
4. The paper proves that GT+QTS achieves roughly the same regret as exact-oracle-based methods
5. The complexity of existing oracle-based approaches grows exponentially with the number of arms, while GT+QTS reduces it to logarithmic complexity
6. The paper is reasonably well organized and well written

Weaknesses

1. The algorithm relies on Assumption 6 (separability of the reward function), which may not hold in practical scenarios, and is only probabilistic (meaning that in cases where the assumption fails, the algorithm could lead to suboptimal selections). While the authors introduce quantized rewards to address this limitation, it is not clear to me how sensitive the method is to different quantization levels
2.  The experimental setup considers only two types of reward functions: Linear rewards and Neural-network-based nonlinear rewards. Although these choices are reasonable, more diverse reward functions (e.g., submodular rewards or real-world application settings) would strengthen the empirical validation.
3. The experiments focus on regret performance, which is great, but it would be useful to also quantify the effect of quantization on arm selection errors
4. The real-world implications of the group testing framework for CMABs are not fully discussed.
5. While the approach is computationally efficient, how would it perform in dynamic environments where reward distributions shift over time? (This does not need to be solved, but at least discussed and it would be nice to have some conjectures or educated guesses on it)
6. The algorithm introduces additional hyperparameters, such as the quantization level $\Delta$ and the probability $p$ of including arms in group tests. More guidance on tuning these parameters would be beneficial.

---

> ### Author Response · Authors · 2025-04-14
> **Thank you for the detailed comments.**
>
> 1. **"Expand on the effect of quantization on arm selection errors (theoretically and/or empirically).":** Thank you for the suggestion. We have added Figure 3(A) to clarify this issue. We have added the following paragraph in Section 5, which we copy below for clarity.
>
> - In order to assess the impact of the quantization interval $\Delta$ on the regret, we perform an ablation study for varying levels of $\Delta$ and its impact on the average cumulative regret. Specifically, we adopt the linear reward setting, i.e., $r(\mathcal{S},\boldsymbol\theta) = \sum{i\in\mathcal{S}} \theta_i$, and we set $m=505$ arms, of which we select a subset of $K=5$ arms in each iteration. We have performed  $50$ independent trials, and the average results of this experiment are provided in Figure 3(A). We observe that as the quantization level increases, the regret increases. This is expected since the larger the gap, the less distinguishable the sub-optimal super-arms are from the optimal ones, resulting in a higher probability of erroneously selecting sub-optimal super-arms. Selecting such super-arms inevitably leads to increased regret.
>
> 2. **"Expand on the real-world implications of the group testing framework for CMABs":** Thanks for the suggestion. We have now included a discussion on the real-world implications of the group testing framework in Appendix A.
>
> 3. **Extension to dynamic environments:** This is a great question. In dynamic environments, the ground truth model $\boldsymbol\mu$ is expected to evolve over time, and as a result, we face a {\em sequence} of models $\{\mu(t) : t\in[T]\}$. In the dynamic setting, an extension of the GT-QTS algorithm has to perform two additional tasks. (1) It should track the underlying ground truth, and, (2) it should potentially estimate the evolving super-arm gaps to decide on the quantization level as well as the number of group tests. Both these steps pose their own challenges. For the first step, we would need to modify the Thompson sampling-based estimator to account for evolving arms in a regret-efficient fashion. For the second step, while one could argue that we assume the knowledge of a *minimum gap* over the sequence of evolving models, a quantization scheme based on this would be highly inefficient, since we can pay a high price (in terms of function evaluations) for models which are potentially easy to find an optimal super-arm. Addressing these challenges requires careful design and is a promising future direction. We will discuss this in the last section. As suggested by the reviewer, we have added the following discussion in Section 6.
>
> - A promising direction is to investigate the impact of using group testing for CMABs in dynamic environments, i.e., when the ground truth model $\boldsymbol\mu$ is expected to evolve over time, and as a result we face a *sequence* of models $\{\mu(t) : t\in[T]\}$. We conjecture that an extension of the GT-QTS algorithm to dynamic environments is highly non-trivial, owing to challenges induced by the estimator (which now has to track an evolving model), and the group testing-based super-arm selection, which crucially hinges on designing an appropriate $\Delta$ – a quantity which is also evolving in the dynamic setting.
>
> 4. **More diverse reward functions:** Thanks for the valuable question. We had chosen the two-layer neural network since it encompasses a rich class of nonlinear functions, for which we have evaluated the performance of our algorithm GT-QTS against CTS in this setting. Submodular functions could be, in principle, another great choice. However, we note that they may not necessarily be separable, which is why we have not provided an experiment involving submodular functions.

---

### Review · Reviewer_PRDS · 2025-03-30

**Summary Of Contributions:**

This paper studies the combinatorial multi-armed bandit (CMAB) problem with semi-bandit feedback and cardinality constraints, addressing the practical challenge that most existing CMAB algorithms assume access to an exact optimization oracle, which may be computationally infeasible. To mitigate this, the authors propose a novel approach that combines group testing for efficient super arm selection with quantization techniques for parameter estimation. Under a probabilistic separability assumption, they design a new Thompson Sampling algorithm that matches the regret bound of the Combinatorial Thompson Sampling (CTS) algorithm by Wang and Chen (2018), while reducing the number of oracle calls from $O(m)$ to $O(\log m)$. Theoretical findings are validated with empirical experiments.

**Audience:**

Yes

**Broader Impact Concerns:**

This is a theoretical work and there is no concerns on the ethical implications of the work.

**Claims And Evidence:**

Yes

**Requested Changes:**

1.	Add discussion on how to estimate $F_{\mu}$ or lower bounds on $\Delta_{\min}$, and include an ablation study on the impact of inaccurate $\Delta$ in the algorithm.
2.	Include experiments on more realistic datasets (e.g., Yelp, Movielens) to better assess performance in practical settings.
3.	Update the related work section to incorporate recent advancements in CMAB, particularly those addressing semi-bandit feedback and using UCB or Thompson Sampling approaches.

**Strengths And Weaknesses:**

**Strengths**

1.	The problem is well-motivated, focusing on reducing the oracle usage to optimize computational efficiency while maintaining regret guarantees.
2.	The application of group testing in the CMAB setting is novel and clever, leading to exponential reductions in oracle complexity without compromising regret bounds.
3.	The paper is clearly written and technically sound, presenting theoretical analysis and empirical validation.

**Weaknesses**

1.	Assumption 5 requires knowledge of the distribution $F_{\mu}$ of the minimum reward gap $\Delta_{\min}$, which is typically unknown in real-world scenarios. Although the authors avoid using this distribution directly in the algorithm and only assume access to its quantile for analysis (e.g., $\Delta = F_{\mu}^{-1}(\gamma)$), it would be valuable to include a discussion on how $F_{\mu}$ (or a lower bound on $\Delta_{\min}$) could be estimated in practice—possibly via heuristics. Moreover, an ablation study evaluating the impact of inaccurate $\Delta$ on regret would make the results more robust and actionable.
2.	Experimental evaluation could be improved. The current synthetic setting feels somewhat artificial. The authors are encouraged to incorporate more realistic datasets, such as Yelp or MovieLens, to better reflect real-world arm distributions and validate the algorithm’s performance in practical applications. Moreover, more baselines should be added such as CUCB [1] and variance-adaptive UCB [2,3].
3.	Related work is missing several recent advances in CMAB with semi-bandit feedback using UCB algoirthms [1, 2, 3, 4] and TS [5] algorithms. The authors should add these new references (but not limited to these) to enhance the timeliness of the work.

[1] Wang, Qinshi, and Wei Chen. "Improving regret bounds for combinatorial semi-bandits with probabilistically triggered arms and its applications." Advances in Neural Information Processing Systems 30 (2017).

[2] Merlis, Nadav, and Shie Mannor. "Batch-size independent regret bounds for the combinatorial multi-armed bandit problem." In Conference on Learning Theory, pp. 2465-2489. PMLR, 2019.

[3] Liu, Xutong, Jinhang Zuo, Siwei Wang, Carlee Joe-Wong, John Lui, and Wei Chen. "Batch-size independent regret bounds for combinatorial semi-bandits with probabilistically triggered arms or independent arms." Advances in Neural Information Processing Systems 35 (2022): 14904-14916.

[4] Merlis, Nadav, and Shie Mannor. "Tight lower bounds for combinatorial multi-armed bandits." In Conference on Learning Theory, pp. 2830-2857. PMLR, 2020.

[5] Perrault, Pierre. "When combinatorial thompson sampling meets approximation regret." Advances in Neural Information Processing Systems 35 (2022): 17639-17651.

---

> ### Author Response · Authors · 2025-04-14
> **Thank you for your detailed comments.**
>
> 1. **"Add discussion on how to estimate $\mathbb{F}_{\boldsymbol\mu}$...":** Thank you for the thoughtful suggestions. We have added the following discussion in Section 2.
>
> - **Lower bounds on $\Delta_{\min}$:** In many practical settings, this is an artifact of designing and representing the experiments by bandit arms. Specifically, in real-world settings, when two experiments are deemed to have close-enough rewards/utilities, they are effectively considered the same. From this perspective, $\Delta_{\min}$ can be considered the minimum separation of rewards based on which we consider the experiments sufficiently distinct that warrants representing them by distinct arms. So, this is an application-specific constant, and depending on the underlying application of interest and what the arms represent, it can be set by the domain expert.
>
> - **Known lower bounds from quantized feedback:** In various applications, feedback or utility values are inherently quantized, leading to a **natural lower bound** on the possible difference between two super-arm utilities. For example, consider the case of a recommendation system where the learner aims to suggest content based on user feedback, balancing exploration (new content) with the avoidance of low-quality recommendations. User feedback in such systems is typically **discrete** (e.g., ratings in the set ${1, 2, 3, 4, 5}$). In this case, the smallest possible difference in utility between any two super-arms is lower bounded by $1$, providing a concrete and known lower bound on $\Delta_{\min}$.
>
> - **Impact of inaccurate $\Delta$:**  In order to assess the impact of the quantization interval $\Delta$ on the regret, we perform an ablation study for varying levels of $\Delta$ and its impact on the average cumulative regret. Specifically, we adopt the linear reward setting, i.e., $r(\mathcal{S},\boldsymbol\theta) = \sum_{i\in\mathcal{S}} \theta_i$, and we set $m=505$ arms, of which we select a subset of $K=5$ arms in each iteration. We have performed  $50$ independent trials, and the average results of this experiment are provided in Figure 3(A). We observe that as the quantization level increases, the regret increases. This is expected since the larger the gap, the less distinguishable the sub-optimal super-arms are from the optimal ones, resulting in a higher probability of erroneously selecting sub-optimal super-arms. Selecting such super-arms inevitably leads to increased regret.
>
> 2. **"Include experiments on more realistic datasets...":** Thanks for the thoughtful suggestion. We have now added a plot with the Movielens dataset in Section 5. Below, we copy the changes in the draft due to incorporating this plot.
>
> - **Real-world datasets:** In order to capture the regret efficiency of the proposed GT-QTS algorithm, we also test its performance on the \texttt{Movielens-100K} dataset, which consists of $100,000$ ratings from 943 users for 1682 movies. Each user is asked to annotate a minimum of 20 movies. In the experiment, we uniformly randomly pick a user and adopt the goal of recommending a set of $5$ movies to match the user’s preference. Here, movies are designed as arms of a multi-armed bandit. In each round, the learner selects a superarm of size $K=5$ and receives feedback (rating) for each arm. The feedback is a Bernoulli random variable with mean value set to the (scaled) original rating from the dataset. Subsequently, the reward corresponding to the superarm is chosen as the sum of feedback from the selected arms. For implementing the GT-QTS algorithm, we set $\Delta=0.2$, which readily follows from the observation that movie ratings lie in the set $\{1,2,3,4,5\}$. In Figure~3(C), we compare the performance of GT-QTS against CTS, which shows a comparable regret performance to the baseline, which assumes access to an exact oracle.
>
> 3. **"Update the related work section...":** Thanks for the additional references; we have updated Section 1 with these.
>
> 4. **"Moreover, more baselines should be added such as CUCB [1] and variance-adaptive UCB [2,3].":** Thanks for the suggestion. However, please note that CTS is state-of-the-art (SOTA), and  [Wang and Chen, 2018]  provides a comparison of the CTS algorithm against CUCB (see Figures 1a and 1b). Those empirical evaluations demonstrate that CTS is **significantly** more regret-efficient compared to CUCB. That is why we use CTS as the SOTA baseline.

---

> > ### Comment · Reviewer_PRDS · 2025-04-16
> >
> > Thank you so much for your responses and it addressed my concerns and the paper is in good shape.

---

### Review · Reviewer_yKyq · 2025-04-01

**Summary Of Contributions:**

This paper presents computationally efficient algorithms for combinatorial semi-bandits in the setting where the number of arms is extremely large. There is a permissible set of super-arms is denoted $\mathcal{I} \subset 2^{[m]}$ with $\mathcal{A} = [m]$, and upon querying this super-arm, the learner receives a noisy sample of the mean parameter of each arm in the set (arm mean parameters are denoted $\boldsymbol{\mu} = \{ \mu_i : i \in [m] \} \in [0,1]^m$), and also a noisy sample of the reward corresponding to that super-arm, which is an implicit function of the mean parameters of the arms, denoted $r ( \mathcal{S}, \boldsymbol{\mu})$. The objective is to find the super-arm which obtains the largest reward denoted $\mathcal{S}^\star (\boldsymbol{\mu})$.

The suboptimality gap of a super-arm $\mathcal{S}$ with respect to a parameter estimate $\boldsymbol{\theta}$ is denoted $\Delta (\mathcal{S}, \boldsymbol{\theta}) = \max_{\mathcal{S}’ \in \mathcal{I}} r(\mathcal{S}’,\boldsymbol{\theta}) - r(\mathcal{S},\boldsymbol{\theta})$. The guarantees in the paper are stated in terms of the minimal and maximal suboptimality gaps, defined as,
\begin{align*}
\Delta_{\min} (\boldsymbol{\theta}) &= \min_{\mathcal{S} \in \mathcal{I} : \Delta (\mathcal{S} , \boldsymbol{\theta}) > 0} \Delta (\mathcal{S} , \boldsymbol{\theta}),
\end{align*}
\begin{align*}
\Delta_{\max} (\boldsymbol{\theta}) &= \max_{\mathcal{S} \in \mathcal{I}} \Delta (\mathcal{S} , \boldsymbol{\theta}),
\end{align*}
Which respectively capture notions of identifiability, and scale for the reward functions.

Existing work such as that of \cite{wang2018thompson} has focused on the optimization oracle which for a given parameter estimate $\boldsymbol{\theta}$ returns an optimal super-arm, $\textsf{Alg} (\boldsymbol{\theta}) \in \arg\max_{\mathcal{S} \in \mathcal{I}} r(\mathcal{S}, \boldsymbol{\theta})$. This requires solving an extremely large combinatorial problem over $\mathcal{I}$, which is typically computationally intractable; there are a plethora of NP / approximation hardness results for solving these problems, typically depending on the structure of $r(\cdot,\boldsymbol{\theta})$ or $\mathcal{I}$. This paper proposes algorithms for this problem in settings where the reward function is structured, and it is possible to circumvent the necessity of such computationally intractable oracles.

The paper essentially imposes four assumptions on the structure of the super-arm,

- **[A1.]** **The reward function $r(\mathcal{S}, \cdot)$ is $M$-bounded and $B$-Lipschitz continuous.** Namely, $r(\cdot, \cdot) \in [0,M]$ pointwise and $|r(\mathcal{S}, \boldsymbol{\theta}) - r(\mathcal{S}, \boldsymbol{\theta}') | \le B \| \boldsymbol{\theta}\_{\mathcal{S}} - \boldsymbol{\theta}'\_{\mathcal{S}} \|\_1$ where the notation $A_{\mathcal{S}}$ indicates restriction to the set $\mathcal{S}$ (this needs to be clarified in the paper).
- **[A2.]** **The reward function is monotone.** Namely,
    \begin{align*}
        \forall \boldsymbol{\theta} \in [0,1]^m, \quad \forall \mathcal{S}\_1 \subseteq \mathcal{S}\_2 \subseteq [m],\ r(\mathcal{S}\_1, \boldsymbol{\theta}) \le r(\mathcal{S}\_2, \boldsymbol{\theta}).
    \end{align*}
    This assumption is only meaningful when $\mathcal{I} \subset 2^{[m]}$, since otherwise, the optimal arm must be $\mathcal{S}^\star = [m]$ and independent of $\boldsymbol{\mu}$.
- **[A3.]** **The CDF of $\Delta_{\min} (\boldsymbol{\mu})$ is known.** The paper does not clarify the source of randomness in this distribution, but upon further reading, it appears that $\boldsymbol{\mu}$ is drawn from a prior, and the CDF is computed with respect to this prior.
- **[A4.]** Lastly, but perhaps most crucially, **the reward function is separable.** For any $\boldsymbol{\theta} \in [0,1]^m$ and any optimal arm $a \in \mathcal{S}^\star (\boldsymbol{\theta})$ and any other arm $a' \not\in \mathcal{S}^\star (\boldsymbol{\theta})$, for any $\mathcal{S} \subseteq [m] \setminus \\{ a,a' \\}$,
    \begin{align} \tag{1}
        r ( \mathcal{S} \cup \\{ a \\} , \boldsymbol{\theta}) - r ( \mathcal{S} \cup \\{ a' \\} , \boldsymbol{\theta}) > 0.
    \end{align}
    This structure means that the optimal super-arm for a given $\boldsymbol{\theta}$, for instance, can be constructed simply by picking the top $K$ out of $m$ arms as measured by the score $r ( \\{ a \\}, \boldsymbol{\theta} )$.

The paper establishes bounds on the regret and the computational complexity of a quantized Thompson sampling + group testing based approach for the problem. The bounds are slightly uninterpretable, but have the important characteristics: the group-testing algorithm only carries out $\log(m)$ tests (i.e., queries of $r (\cdot , \boldsymbol{\theta})$). The computational complexity is still $\Omega(m)$, since it relies on estimating a score across all $m$ arms. The regret scales as $\log(T)$, which is the right scaling for this problem, assuming separability-like conditions.

**Audience:**

Yes

**Broader Impact Concerns:**

This work proposes an algorithm of a theoretical nature and is not likely to raise any major ethical implications.

**Claims And Evidence:**

Yes

**Requested Changes:**

### Additional comments

- The notion $Q(\cdot)$ is reused between the quantization function, as well as the observations in each round. It is probably a good idea to use different notations for the two.
- Theorem 1 should not state ``the average regret conditioned on $\boldsymbol{\mu}$'': it is my understanding that this is a Bayesian setting and the $1-2 \gamma$ success probability also includes events which are functions of $\boldsymbol{\mu}$ drawn from the prior. Eq. (20) in the paper relies on treating $\boldsymbol{\mu}$ as a random variable.
- Fig 2c should label the Y-axis by average cumulative regret, since the cumulative regret cannot be decreasing.

**Strengths And Weaknesses:**

### Major comments

- The paper seems to state the semi-bandit problem with respect the collection of subsets $\mathcal{I}$, which correspond to the legal set of super-arms which can be played. However, neither the algorithm restricts itself to sampling super-arms in this set, nor do the analysis / upper bound effectively depend on this set. The upper bound scales with $\log (2^m |\mathcal{I}|) \in [m, 2m]$. Furthermore, Algorithm 2 in the paper queries super-arms according to the test matrix $\boldsymbol{A}$, which contains randomly sampled rows which are not guaranteed to be the indicators of sets in $\mathcal{I}$; the output of the algorithm is a super-arm which is also not guaranteed to be in $\mathcal{I}$.\
Is it correct to say that the results are essentially for the setting $\mathcal{I} = 2^{[m]}$? This is now confusing to me, since it means that the optimal arm under the monotonicity condition (A2) means that the optimal super-arm must be $[m]$. I think the authors may be indicating that the results are for the setting where the optimal comparator super-arm has size $\le K$, and the algorithm must also finally output a super-arm of size $K$ within each round, however, the reward function can be queried on arbitrary super-arms and parameters when designing an optimization oracle. This distinction needs to be made very clear in the paper. The presentation can also be simplified then, since we may replace $|\mathcal{I}|$ by $\binom{m}{K} \asymp m^K$ in the regret bound.

- The algorithm and its analysis are not surprising. The analysis proceeds in two steps, $(i)$ arguing that Thompson sampling is faithful at estimating the arm means $\boldsymbol{\mu}$, $(ii)$ arguing that the group testing can be used to estimate the optimal arms for the current parameter estimate $\boldsymbol{\theta}$ (cf. Lemma 1). The latter is complicated a little bit by the fact that group testing succeeds only when the reward function is strictly separable (i.e., A4 holds with the reward gap in eq. (1) above, to be at least some constant $C > 0$ rather than just strictly positive). But this is resolved by quantizing the rewards so that the gap is exactly zero (so that two arms are either exchangeable, or are distinguishable), and arguing that the optimal super-arm corresponding to $\boldsymbol{\mu}$ typically does not change (Lemma 2) even under quantization. I am not convinced that there are too many technical challenges in arguing this once quantization is assumed: the analysis is essentially of the form $A + B$ where $A = \{ \text{bound on query complexity of GT} \}$ and $B = \{ \text{bound on regret of TS} \}$, and results of both forms have been established many times in the literature.

- Reward monotonicity is a very strong condition. It essentially means that the top $K$ arms in terms of reward $r(\{a \},\boldsymbol{\theta})$ are in the optimal set $\mathcal{S}^\star (\boldsymbol{\theta})$, so I view the group testing approach as a much quicker way of identifying what these arms must be. It would be more interesting if there were weaker conditions under which such a recovery guarantee were true.

- The paper confounds the computational complexity and regret bounds in the experiments: Fig. 2a/2b argues that the regret of the proposed algorithm is not much worse, while the discussion surrounding Fig. 2c argues that the algorithm's cumulative regret improves gracefully as the number of reward queries / tests grow. The fair comparison to make is to identify how much the regret of the algorithms differ by, when the amount of compute is normalized across algorithms. For instance, in Table 1, the comparison showing that the proposed GT+QTS approach scales better in computational complexity, however the cumulative regret in both cases is not going to be the same, and normalizing the table by the number of horizon steps is somewhat meaningless. The reason I point this out is, by a cursory glance at Table 2, it appears that the amount of time taken by QTS+GT @$H=5000$ is comparable with the amount of time taken by CTS @$H=1000$. However, comparing cumulative regret @$H=5000$ for QTS+GT with that of CTS @$H=1000$ appear very close to each other (Fig. 2c). So the gap in computational complexity, after normalizing for the cumulative regret is much less substantial than Table 1 indicates.

- I am not a huge fan of experiments such as that in Table 1 which plot the runtime of two algorithms, without providing any additional implementation details. It is unclear what to attribute the performance gains to, without these details.

- The writing in the paper is good and the assumptions are stated clearly.

### Summarized review

I like the problem that the paper proposes to study, essentially tackling the computational aspects of combinatorial semi-bandit learning (which often forms the bottleneck over worse statistical / regret guarantees. An algorithm cannot achieve low regret, if it cannot be implemented in the first place). I think the paper can be improved if some of the assumptions such as reward monotonicity can be weakened significantly, in a way where it is less obvious what the interaction between the optimizer (i.e., which returns the optimal super-arm) and the exploration algorithm (Thompson sampling) is. I appreciate that the authors decided to evaluate their algorithm empirically, however the settings are quite toy, and so I am not sure how accurately they reflect practical considerations when extended to more general settings. As such, I think the results may be of some interest to the TMLR community.

---

> ### Author Response · Authors · 2025-04-14
> **Thank you for the detailed comments.**
>
> **Questions about the set of super-arms $\mathcal{I}$:** Thanks for raising a crucial point! Indeed, in the entirety of our presentation, we assume that $\mathcal{I} = 2^{[m]}$. The reviewer is also correct that, based on our assumptions, this would imply that $\mathcal{S}^\star(\boldsymbol\mu) = [m]$. However, we are only permitted to sample up to $K$ arms per round, which also holds for the proposed GT-QTS algorithm. As suggested by the reviewer, we would modify the statement of Theorem 1 with $|\mathcal{I}$ replaced by $2^{[m]}$.
>
> On a side note, we would like to emphasize that this is **not a drawback** of our setting, rather, a simplification for the sake of clarity of presentation. We can readily accommodate a scenario where $\mathcal{I}\subset [m]$. The algorithmic change would be to reject tests that don’t belong to $\mathcal{I}$. This comes at no additional cost since we do not need to perform a function evaluation to identify whether or not our test is contained in the constraint set $\mathcal{I}$. Hence, our results hold in this setting, too.
>
> **"The algorithm and its analysis are not surprising...":** Thank you for explaining a high-level understanding of our analysis technique. We would like to clarify the novel steps in the analysis. First, regarding the component $A$, we don’t think it is intuitive to design the number of group tests. The closest investigation is the one in [28], which finds the sufficient number of group tests in the context of parallel feature selection. This methodology does not directly apply in our setting because of the $C$-separability of the feature scores assumed in their context. As a workaround, we propose the idea of quantization to enable the use of group testing. Here, it is neither clear nor an established result on what the quantization granularity should be. In this context, both Lemmas 1 and 2 are new results. Second, regarding the component $B$, in the analysis of Theorem 1, [7] makes an assumption of access to an exact oracle, and the proof of [7] fails since the analysis of the third term [page 20, 7] breaks down. This necessitates a counterpart analysis of this term, which can be found immediately before the proof of Lemma 4. Finally, as a bonus result, we show that two-layer neural networks satisfy the separability assumption (Appendix D), which, to the best of our knowledge, is also a novel result.
>
> **"Reward monotonicity is a very strong condition.":** Thanks for the valuable remark. While expanding the scope of the reward model is always desirable, in this paper, we have adopted the standard assumption in previous studies on combinatorial bandits, such as [Chen et al., 2016a] (see Assumption 3). We note that various practical reward models, including linear and neural network-based reward models, satisfy this assumption.
>
> **"The paper confounds the computational complexity and regret bounds in the experiments":** Thank you for the thoughtful observation and comment. We certainly agree with the reviewer’s point regarding controlling for the complexity (computational time) and then comparing the performance. To address this and properly provide the comparisons, we have now removed Table 1 and added a new plot, which shows the {\em average normalized regret} by the CTS and GT-QTS algorithms for various values of computation times. Specifically, as suggested by the reviewer, we fix the computation time for each algorithm, find the average cumulative regret incurred by the algorithm in the specified time, and normalize it by the number of rounds executed within the time for a fair comparison. This plot can be found in Figure 3(B).
>
> **"I am not a huge fan of experiments such as that in Table 1":** We have now removed Table 1 and included Figure 4 as per the suggestions of the reviewer. Additionally, we have included specific details on the experimental setup.
>
> **The notion $Q(\cdot)$ is reused...":** Thanks. We have modified the notation for the quantization function, which we now denote by $\xi(\cdot)$.
>
> **"Theorem 1 should not state ``the average regret conditioned on $\boldsymbol\mu$ ''":** We would not call the premise of Theorem 1 a Bayesian setting, since we **do not** average over all instances $\boldsymbol\mu$ to state the regret in Theorem 1. Rather, we only assume a prior distribution on $\boldsymbol\mu$, and more specifically, on the minimum gaps $\Delta_{\min}(\boldsymbol\mu)$. The regret in Theorem 1 is conditioned on a realization $\boldsymbol\mu$ chosen from this prior (which we denote by $\mathbb{F}_{\boldsymbol\mu}$).
>
> **Fig 2c should label the Y-axis by average cumulative regret, since the cumulative regret cannot be decreasing:** In Figure 2(c), we indeed plot the {\em average cumulative regret}. Note that this quantity decreases as we increase the number of tests since the larger the number of tests, the closer the regret performance of the GT-QTS algorithm is to that of CTS.

---

### Decision · Action_Editor_Spri · 2025-05-12

**Recommendation:** Accept as is

**Comment:**

This paper considers the problem of combinatorial multi-armed bandits with semi-bandit feedback and proposes a new, more realistic alternative to the "perfect oracle" assumption by combining a group-testing routine for selecting the super-arms and Thompson sampling (TS) for parameter estimation. Under a general separability assumption on the reward function, the proposed algorithm reduces the complexity of the super-arm-selection oracle to be *logarithmic* in the number of base arms while achieving the same regret order as the state-of-the-art algorithms that use exact oracles.

Overall, all reviewers were positive about this submission, especially highlighting the novelty of the group testing + TS approach in the combinatorial MAB context, the attractive exponential reduction in computational complexity while preserving regret bounds that hold under the perfect oracle assumption, and the high quality of writing and exposition in the paper. Multiple reviewers did critique the initial experimental setup in the paper to be limited -- the authors made a substantial effort to address this concern including running more extensive simulations of computational complexity vs regret and experiments on real-world datasets, which largely addressed these concerns. The authors also made some more minor revisions to discuss some implications of their results and expand on certain assumptions made in the framework (e.g. the separability assumption). One concern that was not addressed by the authors was the perception by one reviewer that the separability/monotonicity assumption on the reward function is rather strong. After the author revision and response, all reviewers are in favor of acceptance. Given all of this, I believe this paper meets the bar for publication at TMLR in its current form.

**Audience:**

Yes, I believe the sub-communities within TMLR that work on online learning, multi-armed bandits, reinforcement learning, and statistical learning theory would be interested in the findings of this paper. In addition, researchers who work in statistical signal processing might be interested in the novel application of group testing to combinatorial multi-armed bandits proposed in this paper.

**Claims And Evidence:**

Yes, all reviewers conducted a detailed evaluation of the paper and found the technical claims to be sound.